# Targeting NUPR1-dependent stress granules formation to induce synthetic lethality in Kras[G12D]-driven tumors

Patricia Santofimia-Castaño [1]✉, Nicolas Fraunhoffer[1,2], Xi Liu[1], Ivan Fernandez Bessone[1], Marina Pasca di Magliano[3], Stephane Audebert [1], Luc Camoin [1], Matias Estaras[1], Manon Brenière[1], Mauro Modesti [1], Gwen Lomberk [4], Raul Urrutia[5], Philippe Soubeyran [1], Jose Luis Neira [6,7] & Juan Iovanna [1,8,9,10]✉

## Abstract

**We find that NUPR1, a stress-associated intrinsically disordered protein, induced droplet formation via liquid–liquid phase separation (LLPS). NUPR1-driven LLPS was crucial for the creation of NUPR1-dependent stress granules (SGs) in pancreatic cancer cells since genetic or pharmacological inhibition by ZZW-115 of NUPR1 activity impeded SGs formation. The Kras[G12D] mutation induced oncogenic stress, NUPR1 overexpression, and promoted SGs development. Notably, enforced NUPR1 expression induced SGs formation independently of mutated Kras[G12D]. Mechanistically, Kras[G12D] expression strengthened sensitivity to NUPR1 inactivation, inducing cell death, activating caspase 3 and releasing LDH. Remarkably, ZZW-115-mediated SG-formation inhibition hampered the development of pancreatic intraepithelial neoplasia (PanINs) in *Pdx1-cre;LSL-Kras[G12D]* (KC) mice. ZZW-115-treatment of KC mice triggered caspase 3 activation, DNA fragmentation, and formation of the apoptotic bodies, leading to cell death, specifically in Kras[G12D]-expressing cells. We further demonstrated that, in developed PanINs, short-term ZZW-115 treatment prevented NUPR1-associated SGs presence. Lastly, a four-week ZZW-115 treatment significantly reduced the number and size of PanINs in KC mice. This study proposes that targeting NUPR1-dependent SGs formation could be a therapeutic approach to induce cell death in Kras[G12D]-dependent tumors.**

**Keywords** Kras; ZZW-115; Synthetic Lethality; Stress Granules; NUPR1
**Subject Categories** Cancer; Digestive System

## Introduction

Pancreatic ductal adenocarcinoma (PDAC) is one of the most lethal cancers, with a low life expectancy (8% at 5 years) (Siegel et al, 2022). PDAC cells are continuously challenged by a hypoxic environment, an increase of reactive oxygen species (ROS) accumulation, a lack of nutrients, and osmotic stress. In addition, expression of oncogenes in otherwise normal cells promotes the activation of a poorly understood process described as "oncogenic stress". All these stressing conditions create a hostile situation for the cell where the processes of transformation, progression and survival of malignant pancreatic cells are conditioned by their ability to develop a robust pro-survival stress response. When the integrated stress response is initiated, different "stress sensors" are induced, coordinating the cellular adaptation to stress (Pakos-Zebrucka et al, 2016). Among these sensors, the serine/threonine kinases promote phosphorylation of eIF2α, a main inductor of stress granules (SGs) formation (McInerney et al, 2005). SGs are conserved cytoplasmic membrane-less organelles (MLO) that contain translation initiation factors, 40S ribosomal subunits, a variety of RNA binding proteins, and many non-RNA binding proteins, which regulate, under stress conditions, mRNA localization, translation, and degradation, as well as signaling pathways (Glauninger et al, 2022). In addition, SGs formation has major advantages for cell physiology since it minimizes energy expenditure, controls protein and ribostasis, and improves cell survival under damaging conditions (Marcelo et al, 2021). Moreover, they have a very dynamic nature, they are quickly assembled under stress and rapidly disperse after the stress situation disappears (Hofmann et al, 2021). However, SGs are heterogeneous MLO structures in which the proteins and RNAs contained can be stress-, cell- and disease-dependent (Jain et al, 2016; Markmiller et al, 2018).

[1]Centre de Recherche en Cancérologie de Marseille (CRCM), INSERM U1068, CNRS UMR 7258, Aix-Marseille Université and Institut Paoli-Calmettes, Parc Scientifique et Technologique de Luminy, 163 Avenue de Luminy, 13288 Marseille, France. [2]Universidad de Buenos Aires, Consejo Nacional de investigaciones Científicas y Técnicas, Centro de Estudios Farmacológicos y Botánicos (CEFYBO), Facultad de Medicina, Buenos Aires, Argentina. [3]Department of Surgery, University of Michigan, Ann Arbor, MI, USA. [4]Division of Research, Department of Surgery, Medical College of Wisconsin, Milwaukee, WI, USA. [5]Genomic Science and Precision Medicine Center (GSPMC), Medical College of Wisconsin, Milwaukee, WI, USA. [6]IDIBE, Universidad Miguel Hernández, Edificio Torregaitán, Avda. del Ferrocarril s/n, 03202 Elche, Alicante, Spain. [7]Instituto de Biocomputación y Física de Sistemas Complejos (BIFI), Universidad de Zaragoza, 50018 Zaragoza, Spain. [8]Equipe Labellisée La Ligue, 2022 Marseille, France. [9]Hospital de Alta Complejidad El Cruce, Florencio Varela, Buenos Aires, Argentina. [10]University Arturo Jauretche, Florencio Varela, Buenos Aires, Argentina. ✉E-mail: patricia.santofimia@inserm.fr; juan.iovanna@inserm.fr

The involvement of RNA granules in cancer initiation and progression is an emerging concept in tumor biology. In this regard, responding and adapting to stress is important in both cancer development and the tumor response to anti-cancer therapies. Recent studies have indicated that several components of SGs participate in tumorigenesis and cancer metastasis (Omer et al, 2020) by, for example, inhibiting cell death (Arimoto et al, 2008). In addition, some chemotherapeutic drugs have been reported to induce SG formation and inhibition of SG-driven proteins, such as G3BP1, and to reduce chemotherapy resistance in several types of cancer (Zhao et al, 2021; Zhang et al, 2019). SGs function as a resistance mechanism to current chemotherapies against PDAC; therefore, interfering with their formation could provide an effective, new approach to sensitizing to chemotherapeutic agents *Kras*-mutated tumors, a mutation found in almost all pancreatic cancer cells. In this respect, targeting proteins involved in the integrated stress response that promote the formation of SGs is gaining considerable interest. However, SGs are heterogeneous MLO structures in which the proteins and RNAs contained can be stress-, cell- and disease-dependent (Jain et al, 2016; Markmiller et al, 2018).

Intrinsically disordered proteins (IDPs) are conformationally flexible, facilitating interactions with multiple partners through intramolecular and intermolecular mechanisms; in addition, IDPs foster a great functional diversity (Oldfield and Dunker 2014). Recent studies showed that low-complexity sequences of IDPs promote liquid–liquid phase separation (LLPS) (Nott et al, 2015; Pak et al, 2016; Toll-Riera et al, 2012; Uversky, 2017), a process that is driven by multivalent protein-protein and/or protein-nucleic acid interactions. Remarkably, LLPS represents a vital and ubiquitous system of intracellular, multi-molecular organization, since it is involved in several physiological and pathological functions underlying the formation of MLO, as the SGs (Wheeler et al, 2016).

Our team has been working for decades on Nuclear Protein 1 (NUPR1), a 82-residue-long IDP that plays an important role in stress response in several tissues (Santofimia-Castaño et al, 2019b). NUPR1 expression is activated in response to some, if not all stresses, including minimal ones (Garcia-Montero et al, 2001). Importantly, NUPR1 was found to be overexpressed in many cancerous tissues, where its expression is essential for their development and progression (Santofimia-Castaño et al, 2019a). In addition, we have described that NUPR1 is involved in pancreatic intraepithelial neoplasia (PanINs) development (Hamidi et al, 2012) and PDAC formation (Cano et al, 2014) in mice, but its mechanism of action remains unexplained. Finally, we have developed a potent NUPR1 inhibitor, named ZZW-115, a small molecule with high anti-cancer activity in vitro and in vivo in several tumor models (Santofimia-Castaño et al, 2019a; Lan et al, 2020).

Given that NUPR1 is a stress-induced IDP and that LLPS is often promoted by IDPs, we hypothesized that expression of NUPR1 could be involved in the SGs formation under several physiological or pathological conditions. In this study, we demonstrate that NUPR1 drives LLPS for SGs development since its pharmacological inactivation is sufficient to block LLPS and SGs formation. In addition, we demonstrate that while mutated *Kras* activation sensitizes cells to SGs formation (Grabocka and Bar-Sagi, 2016), inhibition of the NUPR1-dependent SGs formation with ZZW-115 in *Pdx1-cre;LSL-Kras^G12D^* mice blocks the transformation process, by killing *Kras^G12D^* cells by activating apoptosis, suggesting that SGs formation is necessary for transformation and PDAC development. In addition, short-term treatment of these mice with ZZW-115 blocks the formation of SGs, whereas longer treatment reverses or, at least, strongly retards the development of PanINs, and therefore the development of the PDAC.

# Results

## NUPR1 undergoes LLPS in vitro and interacts with SGs components

There is evidence that IDPs or intrinsically disordered regions (IDRs) of certain proteins can drive LLPS to form droplets in vitro. To determine whether NUPR1 can undergo LLPS on its own, we studied the behavior of recombinant NUPR1 protein (rNUPR1). We prepared solutions of increasing concentrations of wild-type rNUPR1, ranging from 5 to 100 μM, on a buffer (pH 7.2) containing 5% polyethylene glycol (PEG) 8000 and 50 mM NaCl which is a standard experimental condition for evaluating the capacity of an IDP to induce LLPS (Wegmann et al, 2018). Remarkably, rNUPR1 was capable of inducing phase separation in a concentration-dependent manner, as indicated by the formation of droplets (Fig. 1A). This result is not surprising since we have found that at low NUPR1 concentrations there was evidence of high-molecular-weight species, as detected by SAXS, probably due to self-associated NUPR1 species, precursors of droplet formation (Bonucci et al, 2021). We then tested the ability of mutated rNUPR1 at positions A33Q, T68Q or A33Q/T68Q at 50 μM to form droplets. We found that these mutants were almost uncapable of forming LLPS (Fig. 1B). These amino acids were previously reported to be essential to form hydrophobic pockets and interact with protein partners or ZZW-115, a synthetic compound (Santofimia-Castaño et al, 2017). Thus, we added 50 μM of ZZW-115 into the solution of wild-type rNUPR1 and we found an almost complete inhibition of droplet formation (Fig. 1C). Interestingly, the addition of poly ADP-ribose (PAR) at 5 μM, or the addition of 0.2 μg/μl of RNA, both macromolecules involved in LLPS and SGs assembly, to the solution of wild-type rNUPR1 at only 5 μM enhanced its ability to form droplets. However, we did not observe droplet formation with the rNUPR1 mutants or when ZZW-115 was added to the solution (Fig. 1D,E). We also evaluated this droplet-formation capacity under more physiologic conditions by incubating 50 μM of wild-type rNUPR1 in the presence of pure RNA instead of PEG, and we observed LLPS formation (Fig. 1F). These observations demonstrate that NUPR1, but not its mutants, can undergo LLPS in vitro in standard experimental or more physiologic conditions, but not in the presence of the ZZW-115 inhibitor.

We further tested the interaction between wild-type rNUPR1 and either RNA or PAR in vitro by using steady-state fluorescence and far-UV CD spectroscopy. In these experiments, we observed changes in the fluorescence intensity when rNUPR1 formed a complex with either PAR or RNA (Appendix Fig. S1A,C); the changes in fluorescence were larger in the presence of RNA. We also carried out far-UV CD measurements and found that the spectrum obtained from the addition of the spectra of isolated NUPR1 and PAR was different from that of the complex (Appendix

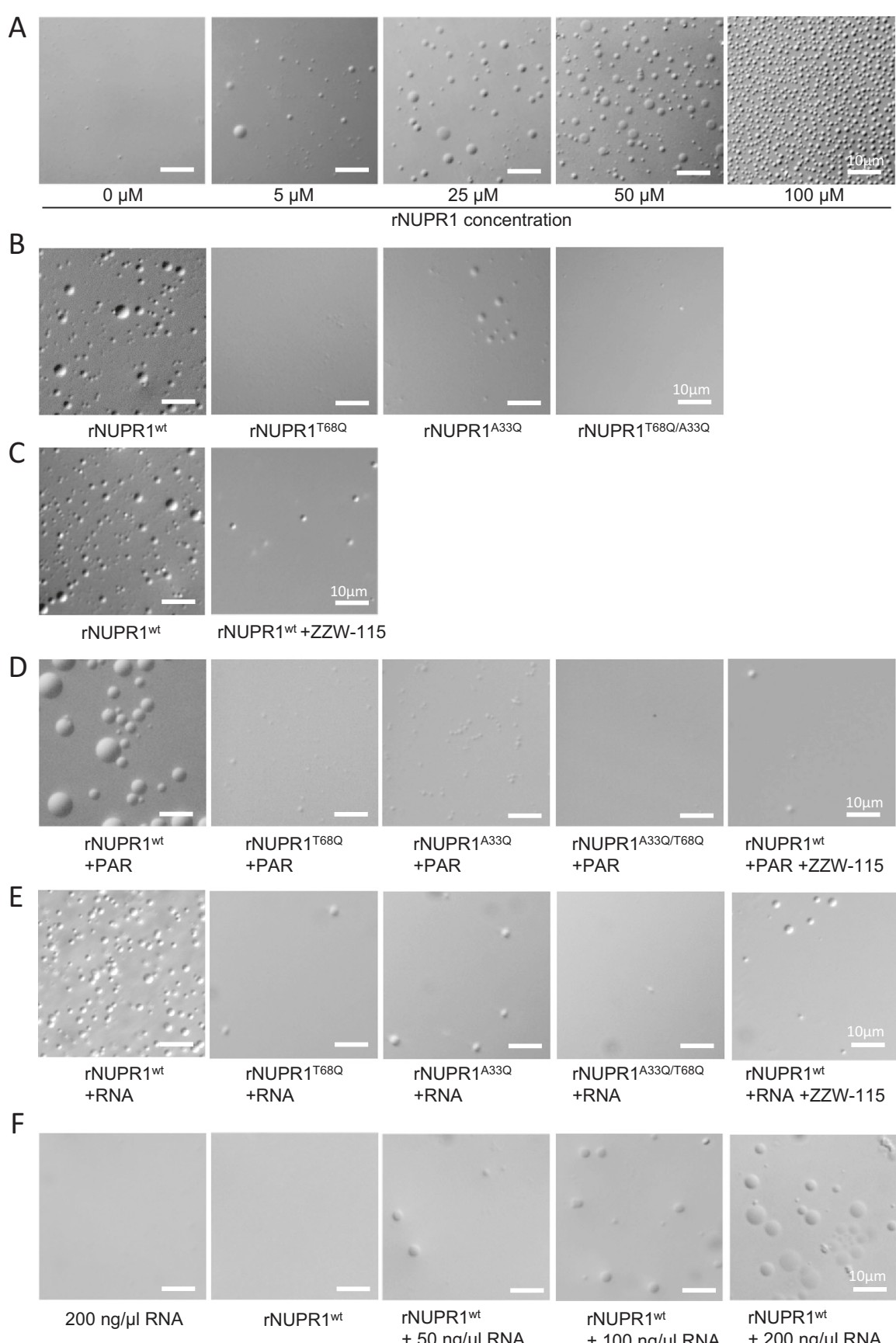

**Figure 1.  NUPR1 undergoes LLPS.**

(A) Differential interference microscopy (DIC) revealed the concentration-dependent formation of liquid droplets of NUPR1 in the presence of PEG-8000 5% and NaCl 50 mM pH 7.2 (Tris buffer). (B) In the same conditions, wild-type rNUPR1 or rNUPR1 mutated on positions A33Q, T68Q, or A33Q/T68Q at 50 μM were tested. (C) The wild-type rNUPR1 at 50 μM was co-incubated with the NUPR1 inhibitor, ZZW-115, at the same concentration. The wild-type rNUPR1 or rNUPR1 mutated on positions A33Q, T68Q, or A33Q/T68Q at 5 μM were co-incubated with (D) Poly(ADP-ribose) (PAR; 5 μM) or (E) RNA (0.2 μg/μl). (F) The wild-type rNUPR1 at 50 μM was co-incubated with RNA at increasing concentration in absence of PEG-8000. Several representative DIC pictures are shown (n = 4). Source data are available online for this figure.

Fig. S1B,D). In the case of RNA and NUPR1, the addition spectrum was also different from that of the complex, but the differences were smaller than those in the far-UV CD spectrum for the PAR/NUPR1 complex. These findings, obtained by using two different spectroscopic techniques, unambiguously indicate binding between NUPR1 and either PAR or RNA. Furthermore, these results show that the binding of NUPR1 to RNA or PAR resulted probably in changes in the secondary structure of NUPR1, as suggested by far-UV CD spectra.

## The stress-inducible protein NUPR1 interacted with proteins involved in SGs formation

The mechanism of SG assembly has been partially understood through proteomic (Jain et al, 2016) and spatial proteomic (Jain et al, 2016; Markmiller et al, 2018; Youn et al, 2018) analysis, which have identified the protein and RNA components of SGs. The central node of this network seems to be G3BP1, which acts as a molecular switch that initiates RNA-dependent LLPS in response to an increase in intracellular free RNA levels. It is further regulated through positive or negative cooperation by external factors that influence the core SG network that comprises approximately 36 proteins and their corresponding associated RNA molecules. However, the nucleic acid and protein composition, as well as the proteins essential for the formation of SGs are not always uniform, since there are G3BP1-independent granules, and they are conditioned by the type of stress (Yang et al, 2020) and likely the cell type. Since our previous results showed that NUPR1 is an IDP stress protein that can undergo LLPS, we hypothesized that this protein could be associated to the SGs formation.

We first analyzed the NUPR1-associated interactome previously generated in our laboratory by co-immunoprecipitation, followed by mass-spectrometry analysis (Santofimia-Castaño et al, 2022) under control and stress conditions such as metabolic or endoplasmic reticulum stresses. We found that the NUPR1 interactome was strongly enriched in proteins involved in RNA metabolism, splicing, and RNA transport. Many of these protein partners are contained within SGs. By immunoprecipitating Flag-tagged NUPR1 under standard cell culture conditions, we observed that NUPR1 interacted with 163 known SG-containing proteins and that these numbers increased to 299 and 218 during glucose starvation and ER stress responses, respectively. These effects were validated by using GFP-tagged NUPR1 as bait (Fig. 2A and Datasets EV 1–7). Since G3BP1 was identified in the co-immunoprecipitation and it is a potent SG-nucleating protein, we confirmed the interaction between NUPR1 and G3BP1 by proximity ligation assay (PLA). As presented in Fig. 2B, a higher number of dots per cell with larger size were observed upon treatment with the best-characterized inducer of SGs, namely inorganic arsenate ($0.226 \pm 0.413$ μm$^3$) *versus* control cells

($0.152 \pm 0.173$ μm$^3$). However, NUPR1-G3BP1 interaction was almost abolished upon treatment with the NUPR1 inhibitor ZZW-115. Thus, these results reveal the existence of NUPR1-dependent SGs.

## SGs formation requires the stress-inducible protein NUPR1

To further study the implications of NUPR1 in SGs formation, we used fluorescence confocal microscopy to monitor NUPR1 and G3BP1 relocation under arsenate treatment. Figure 2D shows that, indeed, in MiaPaCa-2 pancreatic cells arsenate-treatment induced G3BP1 and NUPR1-positive SGs. Therefore, to address whether NUPR1 was also actively involved in SGs formation in pancreatic cancer cells, we inactivated NUPR1 by using a specific siRNA or its pharmacological inhibitor ZZW-115. Treatment of MiaPaCa-2 (Fig. 2C), with either siRNA or ZZW-115 (Fig. 2D), dramatically reduced the formation of SGs. To further support these observations, we induced SGs formation in pancreatic cancer cells obtained from *Pdx1-cre;LSL-Kras$^{G12D}$/INK4a/Arf$^{fl/fl}$/NUPR1$^{+/+}$* or *Pdx1-cre;LSL-Kras$^{G12D}$/INK4a/Arf$^{fl/fl}$/NUPR1$^{-/-}$* mice (Cano et al, 2014). Notably, we found that, compared to control, NUPR1$^{-/-}$ cells were uncapable of forming SGs (Fig. 2E). Control experiments show that genetic inhibition of NUPR1 did not modify the expression of G3BP1 or vice versa (Fig. EV1A). In addition, arsenate or ZZW-115-treatment did not affect the level of G3BP1, but induced a slight increase of NUPR1 expression (Fig. EV1B). Interestingly, overexpression of G3BP1 induced SGs formation only when NUPR1 was expressed. However, in the context of NUPR1 inhibition, by using a siRNA against NUPR1, G3BP1 overexpression did not induce SGs formation (Fig. EV1C). Interestingly, overexpression of G3BP1 induced a reduction in cell viability compared to control (Fig. EV1D). Moreover, when NUPR1 expression was abolished by using a siRNA against NUPR1, these differences in cell growth, induced by overexpression of G3BP1, disappeared. Finally, to address if SGs formation was induced in primary cell lines, with different mutational context, we treated six primary cells derived from a patient with a PDAC with arsenate, and we observed that all of them were capable of forming SGs (Fig. EV1E). In addition, these cells were quite sensitive to ZZW-115, showing a close range of IC$_{50}$ from 3.75 to 5.09 μM (Fig. EV1F). All combined, these results demonstrate that NUPR1 is required in the process of formation of a population of SGs.

## Kras$^{G12D}$ mutation induced NUPR1 overexpression and the formation of NUPR1-dependent SGs

*Kras$^{G12D}$* mutation has been reported to facilitate SGs formation (Grabocka and Bar-Sagi, 2016); thus, our aim was to study the role of *Kras$^{G12D}$*-inducing NUPR1-dependent SGs. To this end, three

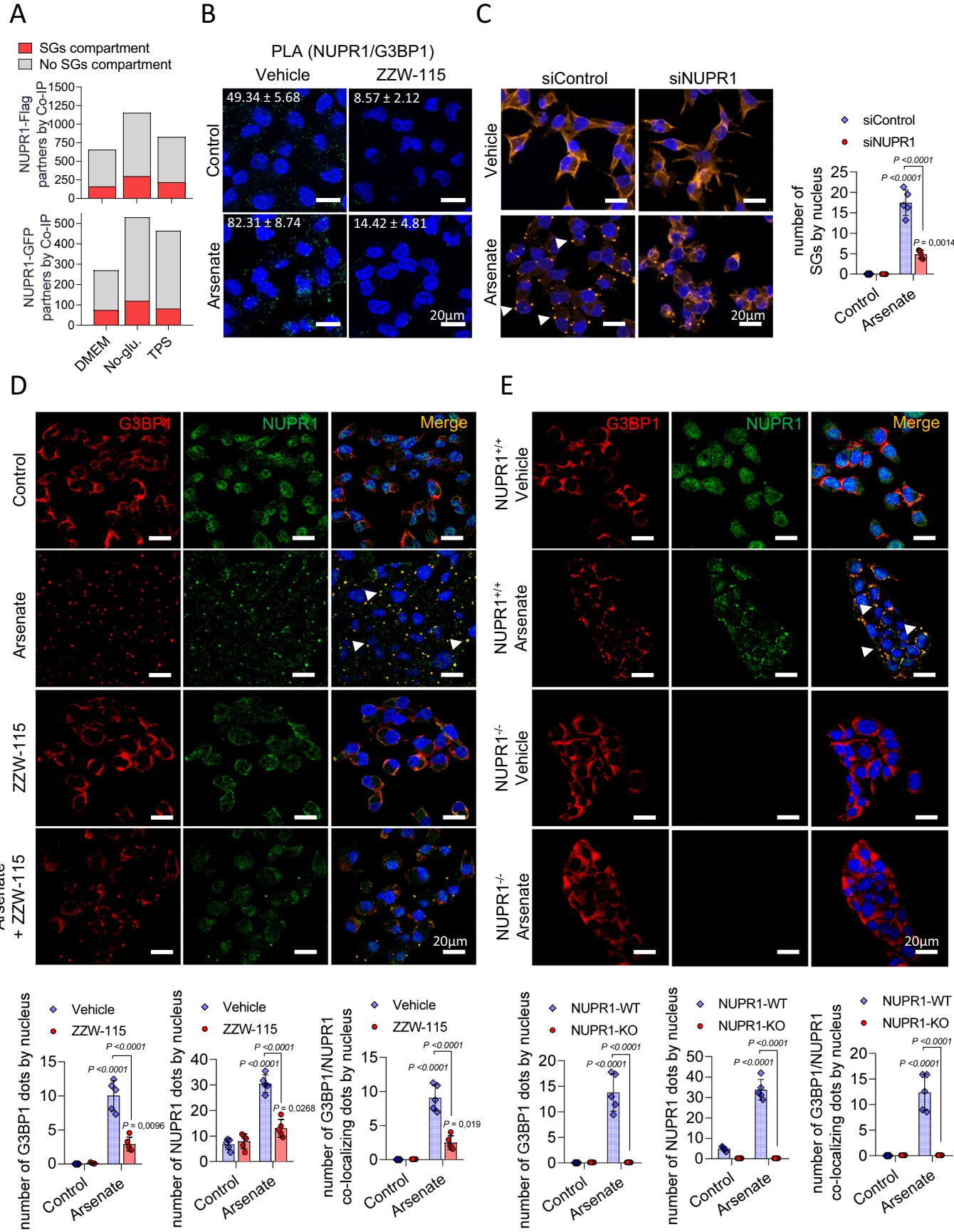

◀ **Figure 2. NUPR1 is essential for the formation of SGs.**

(A) Number of individual proteins identified after co-immunoprecipitation with anti-Flag or anti-GFP agarose beads and LC-MS/MS proteomic analysis in MiaPaCa-2 cells transfected with NUPR1-Flag plasmid (up) or NUPR1-GFP (down). (B) PLA was performed in MiaPaCa-2 cells in the presence or in the absence of ZZW-115 at 6 μM for 6 h in the presence or in the absence of Arsenate at 0.5 mM for 1 h. Mouse anti-G3BP1 and rabbit anti-NUPR1 antibodies were used. A representative experiment is shown (*n* = 4). (C) Immunofluorescence was performed in MiaPaCa-2 cells transfected with siControl or siNUPR1 for 48 h. Mouse anti-G3BP1 and Alexa 568-labeled goat anti-mouse secondary antibodies were used. A representative experiment is shown, arrowheads in the figure highlight the SGs. Quantification of the number of SGs by nucleus is shown (*n* = 5). Data represent mean ± SD, two-way ANOVA with Sidak correction. (D) Immunofluorescence was performed in MiaPaCa-2 cells treated with ZZW-115 at 6 μM for 6 h in the presence or in the absence of arsenate at 0.5 mM for 1 h. Mouse anti-G3BP1 and rabbit anti-NUPR1 and then, Alexa 568-labeled goat anti-mouse and Alexa 488-labeled goat anti-mouse secondary antibodies were used, arrowheads in the figure highlight the SGs. Quantification of number of G3BP1 or NUPR1 by nucleus or number of colocalizing NUPR1/G3BP1 dots by nucleus is shown (*n* = 5). Data represent mean ± SD, two-way ANOVA with Sidak correction. (E) *Pdx1-cre;LSL-Kras^{G12D}/ INK4a/Arf^{fl/fl}/NUPR1^{+/+}* or *Pdx1-cre;LSL-Kras^{G12D}/INK4a/Arf^{fl/fl}/NUPR1^{-/-}* mice cells were treated with Arsenate at 0.5 mM for 1 h. Mouse anti-G3BP1 and rabbit anti-NUPR1 and then, Alexa 568-labeled goat anti-mouse and Alexa 488-labeled goat anti-mouse secondary antibodies were used. Representative pictures are shown, arrowheads in the figure highlight the SGs. Quantification of number of G3BP1 or NUPR1 by nucleus or number of colocalizing NUPR1/G3BP1 dots by nucleus is shown (*n* = 5). Data represent mean ± SD, two-way ANOVA with Sidak correction. Source data are available online for this figure.

pancreatic cancer cell lines derived from a genetically engineered doxycycline-inducible *Kras^{G12D}* transgenic mouse model: 4292 i-*Kras*, 4668 i-*Kras,* and 9805 i-*Kras* cells (Collins et al, 2012; Mathison et al, 2021) were utilized. Doxycycline's possible side-effect was excluded by monitoring metabolic fitness of the cells in the presence of the Kras inhibitor MRTX1133 (Fig. EV2A). First, G3BP1 relocation was monitored, and, as presented in Fig. 3A, induction of *Kras^{G12D}* in these murine pancreatic cell models, significantly increased arsenate-induced G3BP1 relocation as a sign of SGs formation, confirming a previous report (Grabocka and Bar-Sagi, 2016). Since *Kras* mutation activates stress-associated defense processes, we next investigated whether NUPR1, a stress-inducible protein highly upregulated in PDAC cells, was induced by *Kras^{G12D}*. As it can be seen in Fig. 3B, NUPR1 mRNA expression was greater in *Kras^{G12D}* activated cells in the three cell lines. Remarkably, NUPR1 inhibition by siRNA in 4292 i-*Kras*, 4668 i-*Kras* or 9805 i-*Kras* cells almost completely abolished the formation of SGs by arsenate (Fig. EV2B), as well as the inhibition of KRAS signaling by MRTX1133 (Fig. EV2C). It is important to note that while the mutation of the *Kras* induces NUPR1 overexpression, it is not sufficient for the formation of SGs unless the cells are challenged with arsenate, probably because the level of expression is not high enough.

## NUPR1 overexpression triggers NUPR1-dependent SGs formation in pancreatic cancer cells

Given the results obtained above, we hypothesized that upregulation of NUPR1 in pancreatic cells, a phenomenon that occurs in response to various stresses, including *Kras^{G12D}*-induced oncogenic stress, might be responsible of NUPR1-dependent SGs formation. To test this idea, we expressed either NUPR1-Flag wild-type, double mutant NUPR1^{A33Q/T68Q}-Flag or GFP as a control in 9805 i-*Kras* cells. We performed these experiments in the presence or in the absence of doxycycline and arsenate and redistribution of G3BP1 and Flag were studied by confocal microscopy. We found that overexpression of isolated wild-type NUPR1-Flag induced a greater number of SGs, with a co-localization of G3BP1 and wild-type NUPR1-Flag, even in the absence of arsenate. Interestingly, we observed that under doxycycline-minus conditions, cells gained the ability to readily form SGs. On the other hand, when the double mutant NUPR1^{A33Q/T68Q}-Flag or the GFP control constructs were expressed in these pancreatic cancer cells we observed that SGs

cannot be formed (Figs. 3C and EV3A). In addition, when these cells were treated with arsenate, as expected, G3BP1 redistributed and colocalized with wild-type NUPR1-Flag, under both control growth conditions and doxycycline treatment. However, the double mutant acted as a negative dominant of the endogenous NUPR1 since it did not form SGs and therefore, did not induce redistribution of G3BP1 (Figs. 3C and EV3A). Lastly, ZZW-115 in isolation (Fig. EV3B) or ZZW-115 pre-treatment in arsenate-exposed cells (Fig. 3C), inhibited SGs formation in pancreatic cells with forced overexpression of NUPR1, regardless of their exposure to doxycycline. Interestingly, NUPR1 overexpression-derived SGs were not a consequence of an increased G3BP1 expression or *Kras*-signaling, since NUPR1 overexpression of downregulated ERK phosphorylation rather increased it (Fig. EV3C). Altogether these data suggest that a strong overexpression of NUPR1 is sufficient to promote the formation of the NUPR1-dependent SGs population in pancreatic cancer cells, and that its activity can be inhibited by pharmacological means.

## *Kras^{G12D}*-expressing cells are highly sensitive to SGs inhibition

Thus, as *Kras^{G12D}* activation induced NUPR1 overexpression and it played a critical role in the formation of SGs in pancreatic cancer cells, we aimed at analyzing the effect of SGs inhibition in 9805 i-*Kras* cells, upon ZZW-115-treatment or by NUPR1 siRNA. When these cells were analyzed by FACS using Annexin V and propidium iodine staining, we observed that NUPR1 inhibition by ZZW-115 induced cell death in a concentration-dependent manner in cells with the activated *Kras^{G12D}*, while inactivating the expression of this oncogene by removing doxycycline from the media, made these cells resistant to ZZW-115 (Figs. 4A and EV4A). Similar results were observed by using the IncuCyte platform, (Figs. 4B and EV4B), measuring of caspase 3 activity (Figs. 4C and EV4C) and quantification of LDH release (Figs. 4D and EV4D). Interestingly, decrease of caspase 3 activity by Z-VAD-FMK treatment (Fig. EV4E) prevented the cell death induced by ZZW-115 (Fig. EV4F). In this line, inhibition of Kras activity by MRTX1133 also decreased caspase 3 activity induced by ZZW-115-treatment (Fig. EV4G). Remarkably, ZZW-115-treatment reduced cell viability and proliferation independently of *Kras* signaling, since ERK phosphorylation was induced, rather than inhibited, under ZZW-115-treatment (Fig. EV4H). Of note, treatment with

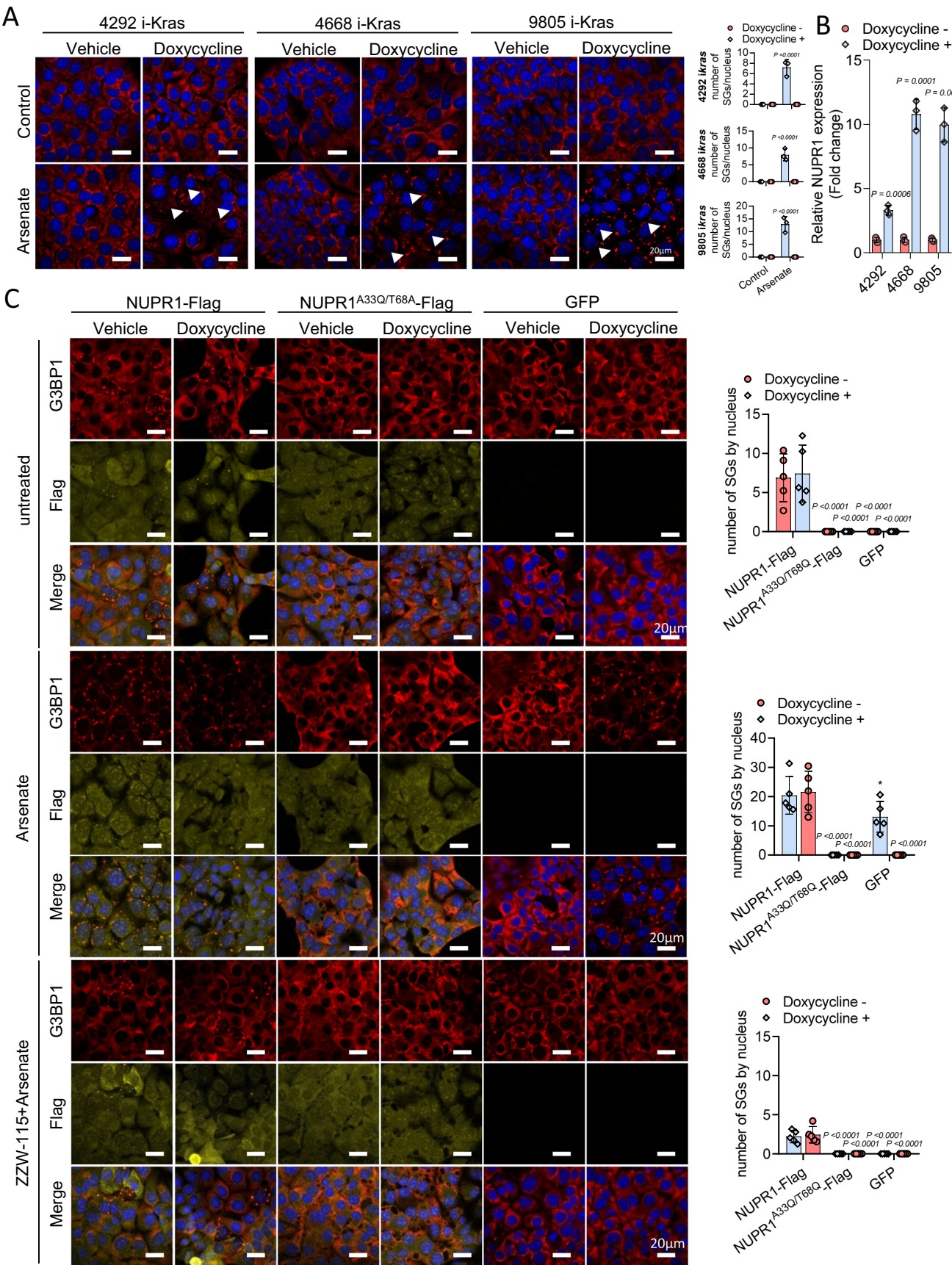

◄ **Figure 3. NUPR1 induced SGs formation in _Kras^{G12D}_-independent manner.**

(A) Immunofluorescence staining was performed in 4292 i-_Kras_, 4668 i-_Kras_ and 9805 i-_Kras_ cells exposed or no to doxycycline and treated with Arsenate at 0.5 mM for 1 h. Mouse anti-G3BP1 and Alexa 568-labeled goat anti-mouse antibodies were used. Representative pictures are shown, arrowheads in the figure highlight the SGs. Quantification of number of SGs in 4292 i-_Kras_ (up), 4668 i-_Kras_ (middle), and 9805 i-_Kras_ cells (down) ($n = 3$, mean of SGs in 5 pictures per n) data represent mean ± SD, two-way ANOVA with Sidak correction. (B) NUPR1 mRNA levels were measured in 4292 i-_Kras_, 4668 i-_Kras_, and 9805 i-_Kras_ cells treated or not with doxycycline, expressed as fold changes ($n = 3$, mean of the fold change, triplicates were done in each independent experiment). Data represent mean ± SD. Student's 2-tailed unpaired $t$ test was used. (C) Immunofluorescence staining was performed in 9805 i-_Kras_ cells 24 h post-transfection of NUPR1-Flag wild-type, its double mutant NUPR1^{A33Q/T68Q}-Flag, or a GFP plasmids in control conditions (up), upon treatment with Arsenate at 0.5 mM for 1 h (middle), or upon treatment with ZZW-115 at 6 μM for 6 h in the presence or in the absence of Arsenate at 0.5 mM for 1 h (down). Mouse anti-G3BP1 and rabbit anti-Flag and then, Alexa 568-labeled goat anti-mouse and Alexa 647-labeled goat anti-rabbit secondary antibodies were used. Quantification of the number of SG by nucleus is shown ($n = 5$). Data represent mean ± SD, two-way ANOVA with Sidak correction. Source data are available online for this figure.

siNUPR1, siG3BP1, or both together, demonstrated that SGs formation was essential for the survival of the oncogenic _Kras_-transformed cells, as shown by cell viability, caspase 3 activity or LDH release results (Figs. 4E–G and EV4I,J,K, respectively). Altogether, these results provide a coherent explanation of the molecular mechanisms underlying the essential role of SGs in _Kras^{G12D}_ mutated cells. It is noteworthy that, in MiaPaCa-2 cells, we detected small NUPR1 + /G3BP1+ dots sensitive to ZZW-115 by using the PLA approach, which could correspond to earlier stress granules (see Fig. 2B). Moreover, we have demonstrated that the inactivation of NUPR1 or G3BP1, two essential factors for the formation of SGs, leads to the death of _Kras^{G12D}_ mutated cells, even in the absence of mature SGs, but only with small dots were present. Furthermore, our data suggest that inhibiting the formation of these MLOs by targeting NUPR1, inhibits cell proliferation only in _Kras^{G12D}_ mutated cells.

## ZZW-115 treatment prevents NUPR1-dependent SG formation and inhibits the PDAC development in _Pdx1-cre;LSL-Kras^{G12D}_ mice

Previous research has shown that NUPR1 is necessary for development of PanINs, an early event in PDAC initiation (Hamidi et al, 2012). In a recent study, Grabocka and Bar-Sagi showed that SGs formation is similarly enhanced in PDAC cells by _Kras^{G12D}_ (Grabocka and Bar-Sagi, 2016) and suggested that SGs may be necessary for oncogenic transformation in vivo, a hypothesis that we decided to further investigate. Consequently, our aim was to understand if NUPR1-dependent SGs were present in the PanINs of _Pdx1-cre;LSL-Kras^{G12D}_ mice and whether the inhibition of this population of SGs could prevent the development of the pancreatic lesions. Thus, we treated _Pdx1-cre;LSL-Kras^{G12D}_ mice and their controls with 5 mg/kg/day of ZZW-115 or a vehicle, once daily, for 10 consecutive weeks starting at 5 weeks of age. It is worth to note that no lesions or NUPR1 expression were found in pancreas of the mice at 5 weeks of age (Fig. EV5A). Upon sacrificing these animals, we confirmed the presence of numerous PanINs in all the _Pdx1-cre;LSL-Kras^{G12D}_ mice (PanINs and ADMs in 8/8 animals), whereas this phenomenon was completely abolished by ZZW-115-treatement (PanINs or ADMs in 0/7 animals) (Figs. 5A and EV5B). This result shows that pharmacological inhibition of NUPR1-dependent SG antagonizes PanINs and ADMs formation in _Pdx1-cre;LSL-Kras^{G12D}_ mice. In addition, staining of pancreatic tissues with the Masson-trichrome technique revealed the deposition of a robust fibrotic tissue only in the _Pdx1-cre;LSL-Kras^{G12D}_ mice (Fig. 5B).

We also investigated whether cells expressing mutant _Kras^{G12D}_ remained present in the pancreas of _Pdx1-cre;LSL-Kras^{G12D}_ mice treated with ZZW-115. For this purpose, we evaluated the expression of phosphorylated-ERK (p-ERK), phosphorylated-AKT (p-AKT), _Kras^{G12D}_, CK-19 and amylase by immunofluorescence. As presented in Fig. 5C–F, activation of p-ERK, p-AKT, KRAS^{G12D}, and CK-19 was observed in tissue sections from _Pdx1-cre;LSL-Kras^{G12D}_ mice, while these signals were not detected in the pancreata of ZZW-115-treated or control animals. Simultaneously, we observed a substantial decrease in amylase expression in the PanINs and Acinar-to-Ductal Metaplasia (ADMs) (Fig. 5F) in the pancreas of the _Pdx1-cre;LSL-Kras^{G12D}_ mice, but amylase remained at the same level as in the control pancreas when these mice were treated with ZZW-115 (Fig. 5G). These results demonstrated that _Kras^{G12D}_ signals were typically activated in this pancreatic cancer initiation model; however, ZZW-115-treatement prevented the presence of cells expressing mutant _Kras^{G12D}_. A western-blot validating the loss of cells expressing p-ERK is presented in Fig. 5H.

The next step was to investigate whether the NUPR1-positive SGs were present in the pancreas of the _Pdx1-cre;LSL-Kras^{G12D}_ mice and if the treatment with ZZW-115 prevented the formation of these structures. By monitoring the localization and distribution of SGs markers G3BP1, phosphorylated-EIF2α (p-ElF2 α), and poly adenine binding protein (PABP), we found that SGs readily formed in the PanINs lesions of _Pdx1-cre;LSL-Kras^{G12D}_ mice. On the contrary, ZZW-115-treatment prevented SGs formation in these mice. We observed that G3BP1 colocalized with NUPR1 in SGs present in the cytoplasm of the cells that formed PanINs lesions in _Pdx1-cre;LSL-Kras^{G12D}_ mice, but not in the pancreas of ZZW-115-treated mice or their control counterparts (Fig. 6A). Similarly, p-EIF2α and PABP colocalized in the pancreas of the _Pdx1-cre;LSL-Kras^{G12D}_ mice within the SGs, while its localization remained unchanged in the pancreas of the animal treated with ZZW-115 and in control mice (Fig. 6B). Altogether, these results demonstrated that NUPR1-dependent SGs were developed in cells from PanINs lesions, but not in ZZW-115-treated pancreatic cells.

However, one last question remained unanswered, what happens to the cells in which the oncogenic _Kras_ is activated during treatment with ZZW-115? We hypothesized that these cells undergoing transformation, were uncapable of bypassing the "oncogenic stress" induced by _Kras_ mutation, and as a result, they failed to survive and ultimately died. To test this hypothesis, we analyzed the expression of mutant Kras^{G12D} in the pancreas from _Pdx1-Cre;Kras^{G12D}_ animals by using immunofluorescence with a specific antibody targeting the mutation, along with the expression of activated caspase 3; simultaneously, we labeled the

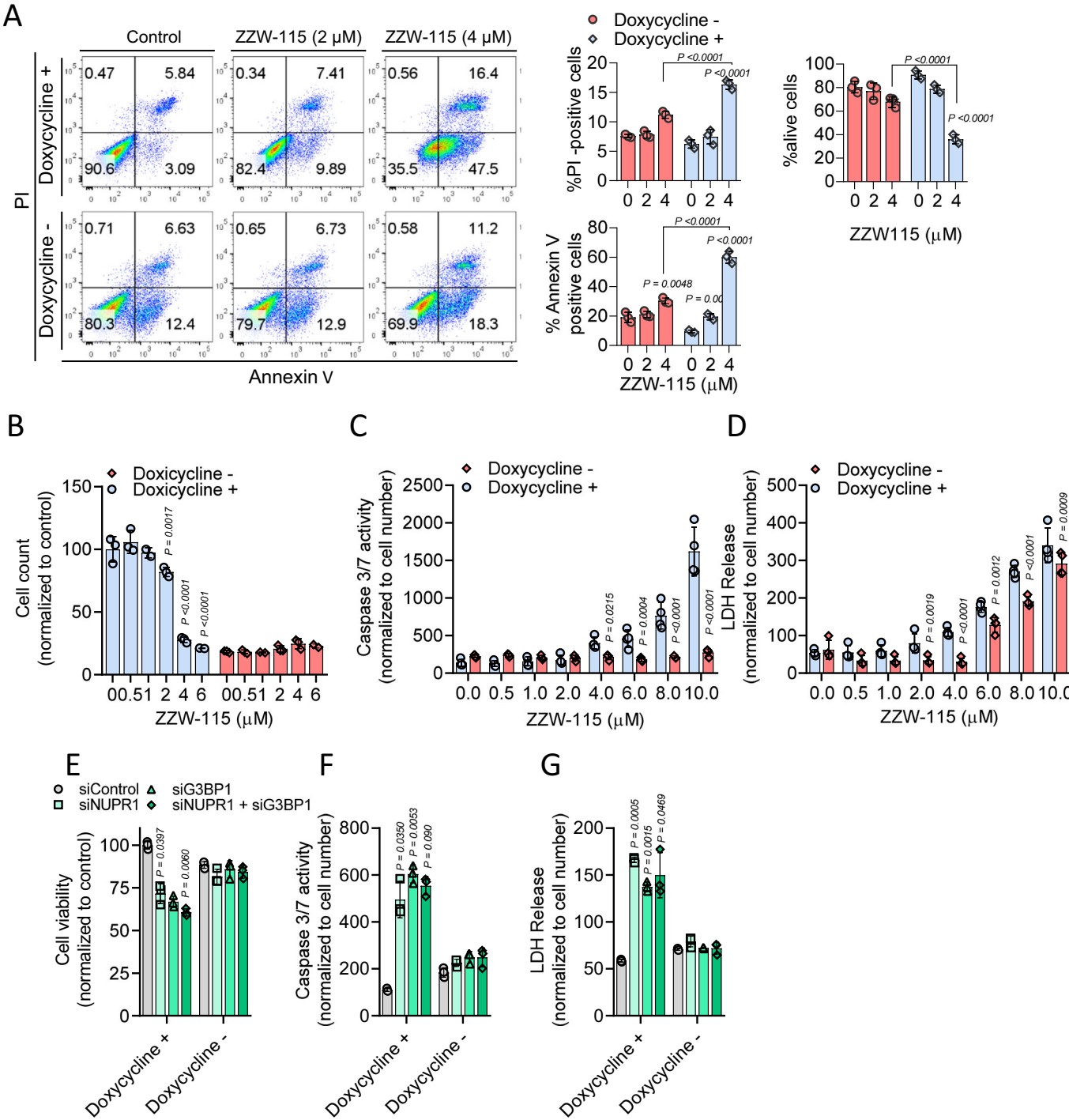

**Figure 4. Inhibition of NUPR1 induced cell death only in *Kras^G12D* cells.**

(**A**) Flow cytometry analysis of annexin V/PI staining was done in 9805 i-*Kras* cells following 24 h of treatment with increasing concentrations of ZZW-115 in presence or not of doxycycline. A representative experiment of the dot plot profile of cells is shown (*n* = 3 independent experiments, triplicates were made on each one). Percentage of PI positive cells, Annexin V positive cells and alive cells were counted. Data represent mean ± SD, two-way ANOVA with Sidak correction. (**B**) Cell count of 9805 i-*Kras* cells measured by IncuCyte live-cell imaging after 30 h of treatment of increasing concentrations of ZZW-115 in presence or not of doxycycline was evaluated (*n* = 3). Data represent mean ± SD, two-way ANOVA with Sidak correction. 9805 i-*Kras* cells were incubated at increasing concentrations ZZW-115 in presence or absence of doxycycline for 24 h and (**C**) caspase 3/7 activity and (**D**) LDH release were measured (*n* = 4) data represent mean ± SD, two-way ANOVA with Sidak correction. (**E**) Cell viability, (**F**) caspase 3/7 activity, and (**G**) LDH release were measured in 9805 i-*Kras* cells transfected with siControl, siG3BP1, siNupr1 or both together for 48 h in the presence or absence of doxycycline (*n* = 3 independent experiments, triplicates were made on each one). Data represent mean ± SD, two-way ANOVA with Sidak correction. Source data are available online for this figure.

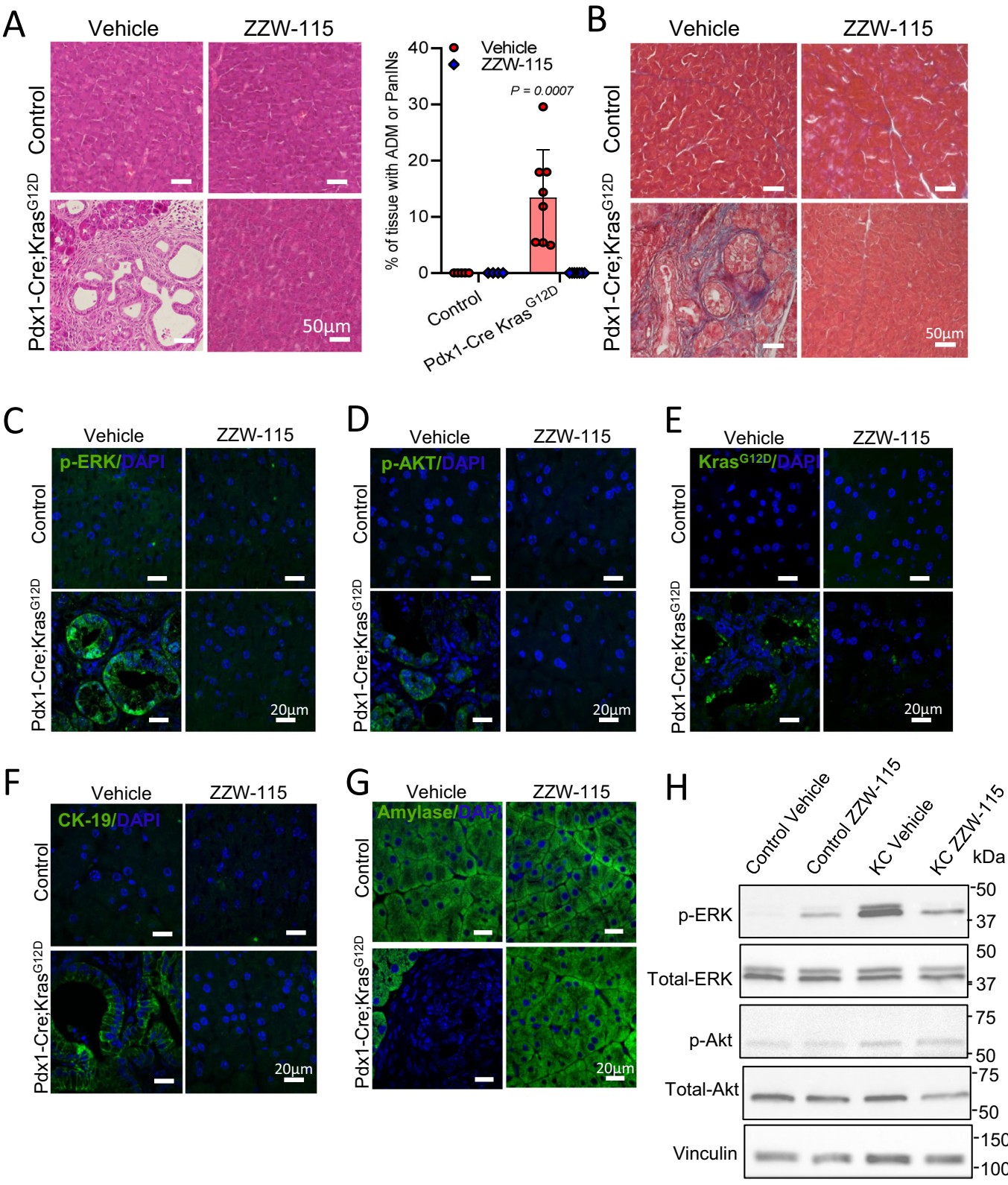

**Figure 5.  NUPR1's in the presence of ZZW-115 treatment inhibits PanINs formation in *Kras^G12D^* mice.**

Representative pictures of histologic sections of the pancreas of control mice or *Pdx1-cre;LSL-Kras^G12D^* mice (both from vehicle or 5 mg/kg ZZW-115-treated mice) stained with H&E (**A**) or Masson-trichrome staining (**B**) (*n* = 3). Quantification of surface with lesion tumors was calculated by ImageJ analysis, data represent mean ± SD, two-way ANOVA with Sidak correction. Immunohistofluorescence staining were performed in histologic sections of the pancreas of control mice or *Pdx1-cre;LSL-Kras^G12D^* mice (both from vehicle or 5 mg/kg ZZW-115-treated mice). Rabbit anti-pERK (**C**), rabbit anti-p-AKT (**D**), rabbit anti-Kras^G12D^ (**E**) or, rabbit anti-CK19 (**F**) or rabbit anti-amylase (**G**) primary antibodies were used, then, Alexa 488-labeled goat anti-rabbit secondary antibody was used (*n* = 3). (**H**) Western-blot analysis was performed to evaluate p-ERK, ERK, p-AKT, AKT, and vinculin levels (*n* = 3). Source data are available online for this figure.

DNA. Figure 7 shows that, as expected, control animals did not exhibit any expression of Kras^G12D^ or activated caspase 3. In contrast, the pancreas of *Pdx1-Cre;Kras^G12D^* animals treated with the vehicle showed strong expression of the mutant form of *Kras*, but no signal of activated caspase 3. Finally, in the pancreas of *Pdx1-Cre;Kras^G12D^* animals treated with ZZW-115, only a few cells expressing the mutated form of *Kras* were detected, but systematically these cells exhibited activation of caspase 3, along with signs of DNA fragmentation consistent with features of apoptotic bodies. From the results of these experiments, we can conclude that the inactivation of NUPR1 by ZZW-115 treatment promotes apoptotic cell death in pancreatic cells expressing the mutated form of *Kras*, similarly to our previous experiments in vitro.

We then monitored the protein levels of KRAS^G12D^, ERK1/2 and phosphorylated-ERK1/2 (p-ERK1/2) as well as AKT and p-AKT by western-blot using specific antibodies in extracts of cells derived from *Pdx1-cre;LSL-Kras^G12D^/INK4a/Arf^fl/fl^/NUPR1^+/+^* or *Pdx1-cre;LSL-Kras^G12D^/INK4a/Arf^fl/fl^/NUPR1^-/-^* mice (Cano et al, 2014) as a possible role of NUPR1 in *Kras*^G12D^ signal transduction. Figure EV5C demonstrates that NUPR1 inactivation did not modify the expression of these proteins. Therefore, resistance to PanIN development in response to *Kras*^G12D^ cannot be explained by inhibition of *Kras* signaling.

### ZZW-115 treatment blocks the NUPR1-dependent SGs formation in PanINs of *Pdx1-cre;LSL-Kras^G12D^* mice

NUPR1 is an essential element for the formation of the SGs containing NUPR1, and these SGs are crucial for PanIN development. Consequently, inhibiting NUPR1-dependent SG formation and, therefore PanINs lesions, could be used as a therapeutic strategy to block their progression toward PDAC. Therefore, the next step was to investigate whether a brief treatment with ZZW-115 in 15-week-old *Pdx1-cre;LSL-Kras^G12D^* mice, which had already developed PanINs, was capable of reversing NUPR1-dependent SGs formation and, consequently, the progression of PanINs. To this end, *Pdx1-cre;LSL-Kras^G12D^* mice were treated with 5 mg/kg/day of ZZW-115 or the vehicle, once daily, for 7 consecutive days starting at 14 weeks of age. Upon sacrificing these animals, we confirmed the presence of numerous PanINs in all *Pdx1-cre;LSL-Kras^G12D^* mice. To quantify the population of SGs in these lesions, we performed immunofluorescence with antibodies against G3BP1 and NUPR1 and their expression was analyzed by confocal microscopy. Remarkably, mice treated with vehicle presented several PanINs with abundant SG positive for both G3BP1 and NUPR1. On the contrary, in mice treated with ZZW-115, although PanINs remained present, the number of SGs existing in the PanINs was significantly decreased (Fig. 8). We concluded that inhibition of NUPR1 by a short

treatment with ZZW-115 can block the SGs development in PanINs.

### ZZW-115 treatment decreased PanINs expansion in *Pdx1-cre;LSL-Kras^G12D^* mice

In our previous experiments, we observed a significant reduction of NUPR1-dependent SGs development following a brief treatment with ZZW-115 in *Pdx1-cre;LSL-Kras^G12D^* mice carrying PanINs. However, our subsequent objective was to investigate whether inhibiting NUPR1-dependent SGs could impede the expansion and progression of PanINs in the pancreas after their initial development. To achieve this objective, we administered a daily treatment of 5 mg/kg/day of ZZW-115 or the vehicle for a duration of four weeks to 14-week-old *Pdx1-cre;LSL-Kras^G12D^* mice with established PanINs, as well as to control mice. After the treatment period, we conducted histological evaluations of the mice's pancreata. As anticipated, the *Pdx1-cre;LSL-Kras^G12D^* mice treated with the vehicle displayed extensive PanINs development. Conversely, those treated with ZZW-115 showed a significant prevention of PanINs expansion after the four-week treatment period (Figs. 9A,B and EV5D). In addition, a higher number of cells expressing cleaved caspase-3 were found in the pancreas of the ZZW-115-treated mice (Fig. 9C). Consequently, our previous findings indicated that the use of ZZW-115 effectively prevented SGs formation in in vivo animal models. As reported by us and others, SGs are present in PanINs and serve as a mechanism for stress defense during cellular transformation. Thus, our current observations demonstrate that inhibiting NUPR1-dependent SGs impedes PanINs expansion.

## Discussion

*Kras* mutations are present in approximately one-third of human tumors, and then they are the most common gene mutations associated with human cancers. It is found in 90% of PDAC, 40% of colorectal cancers, and 32% of lung cancer (Pylayeva-Gupta et al, 2011). In cancer cells, *Kras* mutations induce constitutive oncogenic activation, stimulating cell proliferation, suppressing apoptosis, altering cell metabolism, inducing autophagy, changing cell motility and invasion, and modulating the tumor microenvironment (Ferreira et al, 2022). Two main strategies to target mutant RAS proteins are envisioned: on the one hand, inhibiting the mutated protein or its downstream effectors (Shapiro et al, 2020); on the other hand, identifying therapeutic vulnerabilities in tumor cells addicted to this oncogene (Aguirre and Hahn, 2018; Roman et al, 2022).

Thus, rather than directly targeting oncogenic *Kras* mutations, an original strategy to induce synthetic lethality in *Kras* mutant

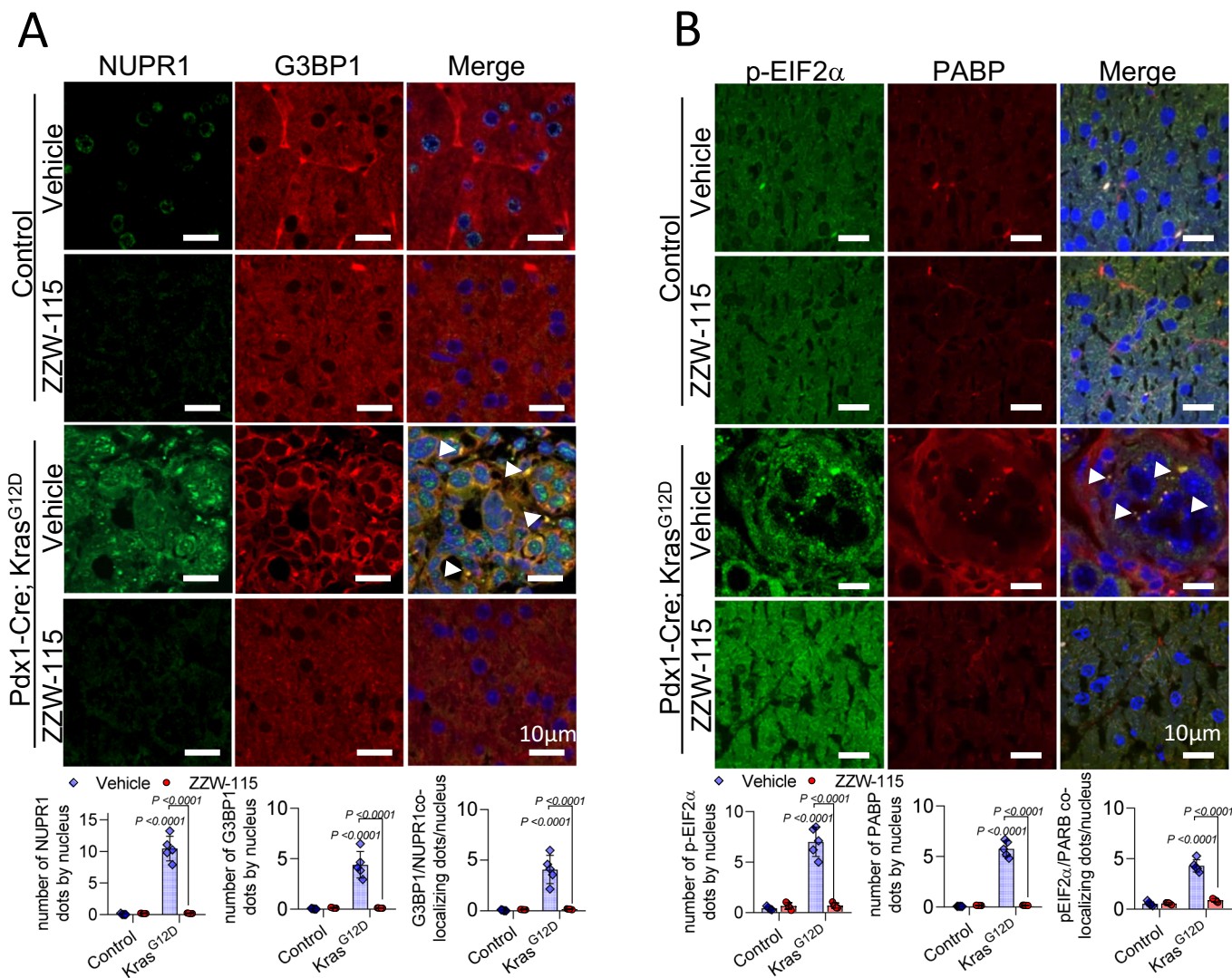

**Figure 6. NUPR1 inhibition prevents PanINs formation and SG development in the pancreas of *Kras^{G12D}* mice.**

Immunohistofluorescence staining was performed on histologic sections of the pancreas of the different experimental groups. Mouse anti-G3BP1 and rabbit anti-NUPR1 (**A**) or Mouse anti-PABP and rabbit anti-p-EIF2α (**B**) and then, Alexa 568-labeled goat anti-mouse and Alexa 488-labeled goat anti-rabbit secondary antibodies were used. Representative pictures are shown, arrowheads in the figure highlight the SGs. Quantification of number of G3BP1, NUPR1, p-EIF2α or PABP by nucleus or number of colocalizing dots by nucleus is shown. Data represent mean ± SD, two-way ANOVA with Sidak correction ($n = 5$). Source data are available online for this figure.

tumors might be to target the oncogenic induced stress by *Kras* mutation. Following this guiding principle, SGs formation in response to activation of mutated *Kras* signaling, as part of the oncogenic stress response, could be considered as an additional promising therapeutic target for synthetic lethality. G3BP1, a key protein involved in SGs development and in tumor progression by regulating the RAS, TGF-β/Smad, Src/FAK and p53 signaling pathways (Zhang et al, 2019), has been considered as a promising target. However, G3BP1 inhibition leads to neonatal death, premature aging phenotype as well as the development of pathologies such as ataxia (Omer et al, 2020). Thus, critical pharmacological investigations must be done to achieve this goal.

Notably, the results reported here extend the importance of this concept, by revealing that NUPR1 plays a crucial role in the formation of, at least, a population of SGs in response to oncogenic

stress induced by the *Kras^{G12D}* mutation. In previous works we demonstrated that *Nupr1* is important for PanINs (Hamidi et al, 2012) and PDAC (Cano et al, 2014) development in mice, since its genetic inactivation resulted in absence of PanINs development in *Pdx1-cre;LSL-Kras^{G12D}* mice and in the reduction of around of 50% of the PDAC development in *Pdx1-cre;LSL-Kras^{G12D}/INK4a/Arf^{fl/fl}* mice. As many other stresses, oncogenic stress induced by the *Kras^{G12D}* mutation also induces NUPR1 overexpression, which in turn facilitates the formation of the NUPR1-dependent SGs, a key mechanism to fight against the oncogenic stress. Indeed, we reveal in this study that NUPR1 promotes the formation of these MLOs likely through its ability to undergo LLPS. Furthermore, we demonstrate that NUPR1 overexpression is sufficient to trigger the formation of NUPR1-contaning SGs formation in cells, highlighting a central direct role of this protein in the development

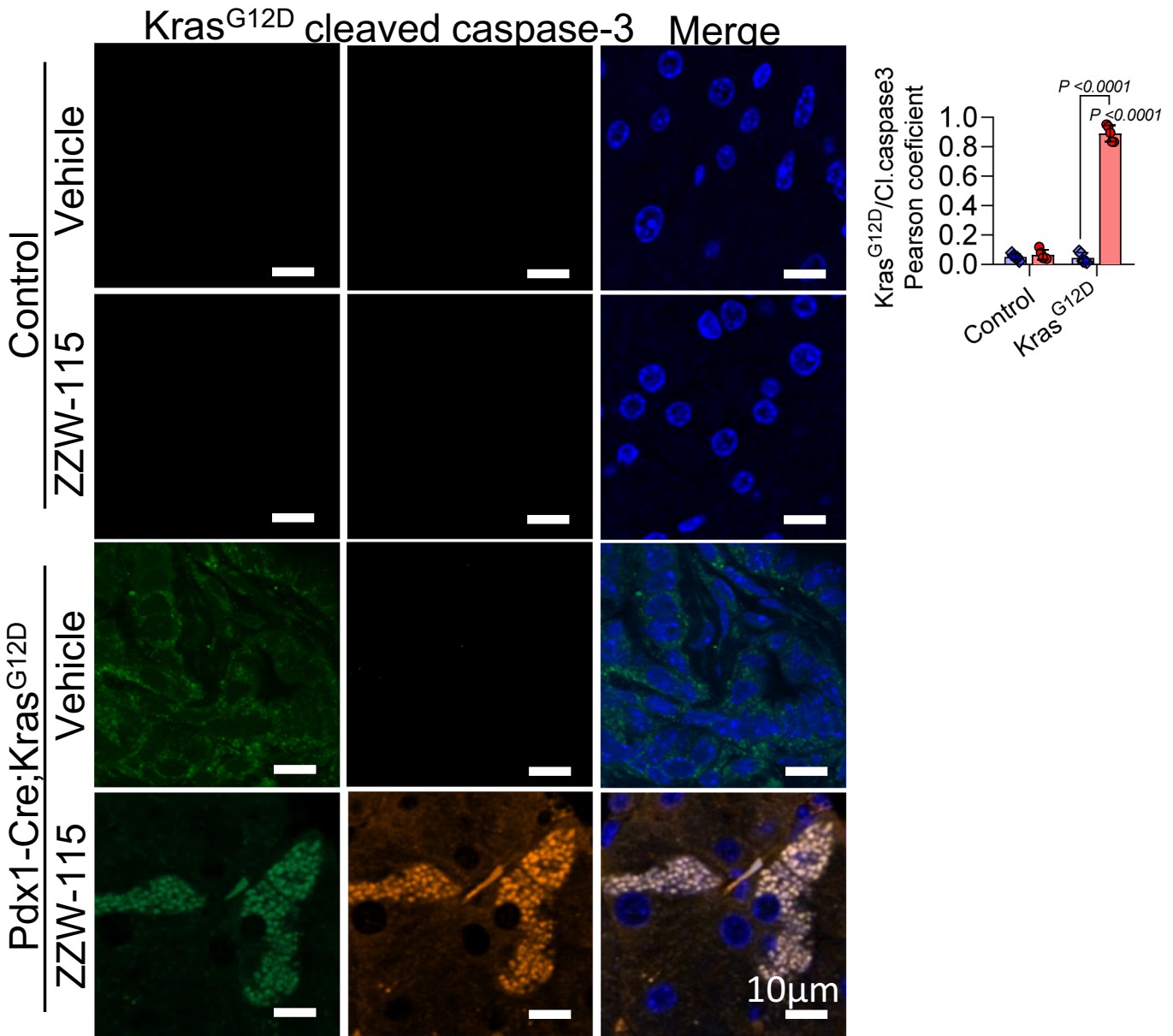

**Figure 7. NUPR1 inhibition induced cell death by apoptosis in *Kras*^G12D expressing cells in vivo.**

Immunohistofluorescence staining was performed on histologic sections of the pancreas of the different experimental groups. Rabbit anti-*Kras*^G12D primary antibody and mouse anti-cleaved caspase 3 antibody was used. DNA was stained with DAPI. Pearson coefficient between both channels was calculated by JACoP, ImageJ. Data represent mean ± SD, ANOVA with Sidak correction ($n = 5$ mice). Source data are available online for this figure.

of these structures. Importantly, we show that inhibiting NUPR1 activity, by using the drug ZZW-115, prevents NUPR1-dependent SGs formation in vitro and in vivo, suggesting that the same polypeptide regions involved in binding to the drug are crucial in forming the SGs. More importantly, inhibition of the formation of SGs is sufficient to either inhibit the development of PanINs or prevent the expansion of them in the pancreas of *Pdx1-cre;LSL-Kras*^G12D mice. This finding supports the concept that counteracting the function of this IDP abolishes these phenomena. Together our findings suggest that targeting NUPR1-dependent SGs formation could be a promising therapeutic strategy for mutated *Kras*^G12D-

associated tumors and provide a preclinical proof-of-concept in support of this approach.

Our main inquiry was to understand why cells presenting SGs were absent in non-transformed areas of the pancreas contrary to PanINs in *Pdx1-cre;LSL-Kras*^G12D mice. We evaluated two possibilities: the first one was that ZZW-115, acting as an inhibitor of the NUPR1, decreased the signaling of the mutated *Kras*^G12D, and consequently its transformation activity or capacity to induce the oncogenic stress response; and the second one was that formation of SGs, induced by overexpression of NUPR1, was essential for transformation activity of the *Kras*^G12D. The first possibility was

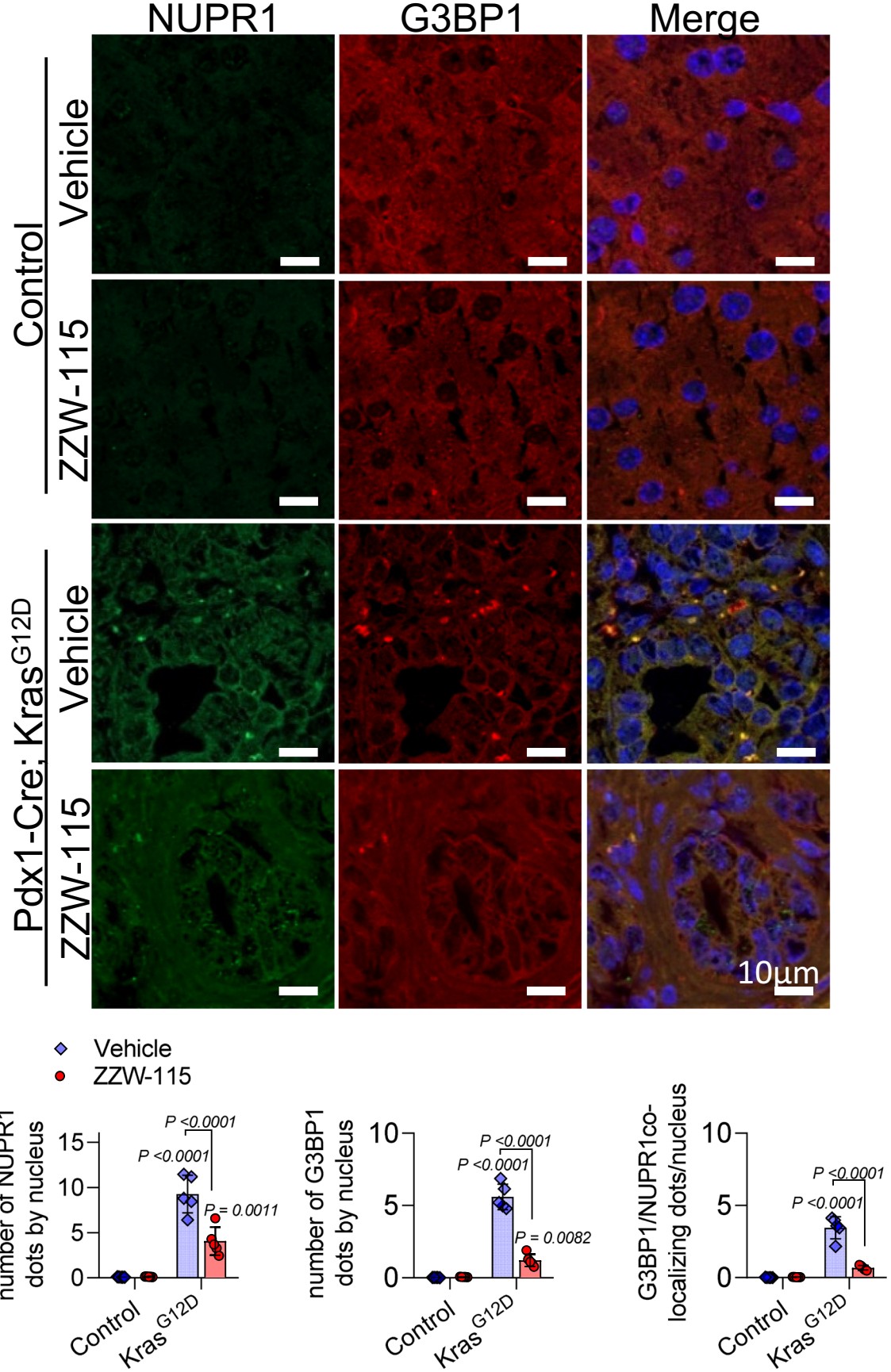

◄ **Figure 8.  NUPR1 inhibition prevents SG formation in the PanINs of *Kras^G12D* mice.**

Immunohistofluorescence staining was performed on histologic sections of the pancreas of control mice or *Pdx1-cre;LSL-Kras^G12D* mice (both from vehicle or 5 mg/kg ZZW-115-treated mice for 1 week starting from the week 14. Mouse anti-G3BP1 and rabbit anti-NUPR1 and then, Alexa 568-labeled goat anti-mouse and Alexa 488-labeled goat anti-rabbit secondary antibodies were used. Representative pictures are shown. Quantification of number of G3BP1 or NUPR1 by nucleus or number of colocalizing NUPR1/G3BP1 dots by nucleus is shown (*n* = 5). Data represent mean ± SD, two-way ANOVA with Sidak correction. Source data are available online for this figure.

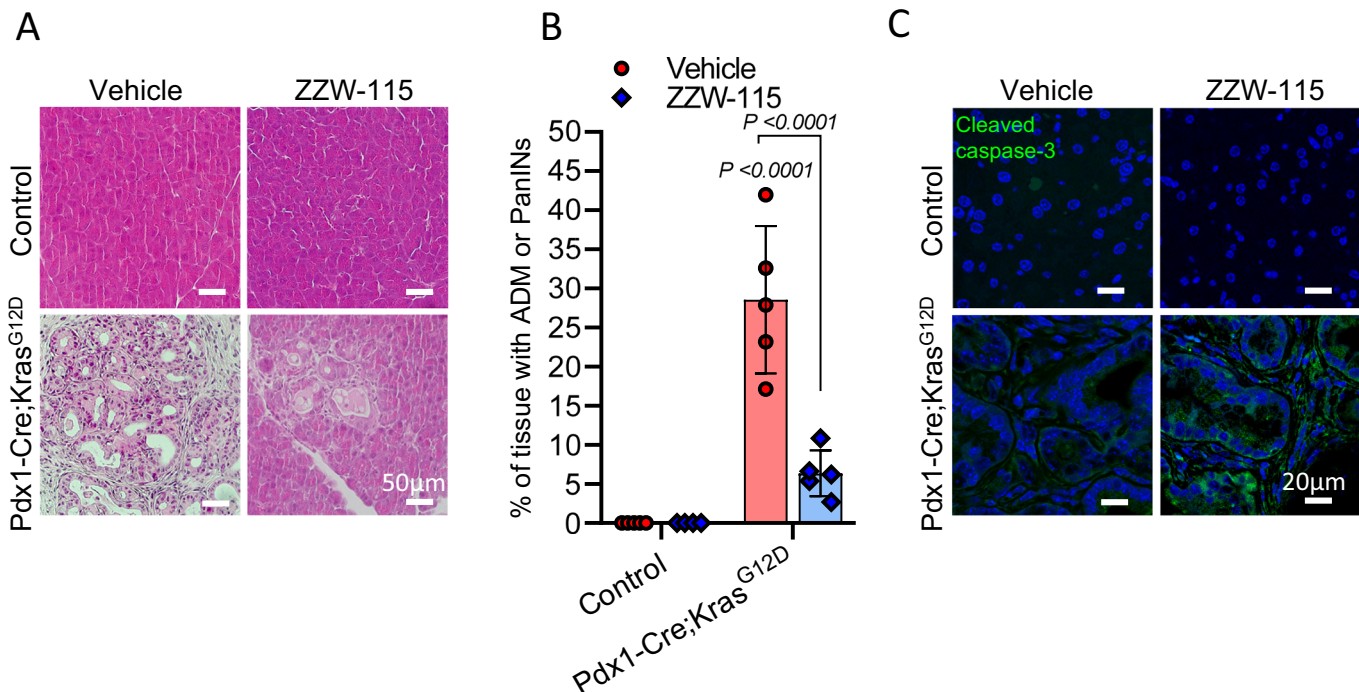

**Figure 9.  NUPR1 inhibition prevents PanINs expansion in the pancreas of *Kras^G12D* mice.**

(A) Representative pictures of histologic sections of the pancreas of control mice or *Pdx1-cre;LSL-Kras^G12D* mice (both from vehicle or 5 mg/kg ZZW-115-treated mice for four weeks starting from the week 14) stained with H&E (*n* = 5). (B) Quantification of surface with lesion tumors was calculated by ImageJ analysis, data represent mean ± SD (*n* = 5), two-way ANOVA with Sidak correction. (C) Immunohistofluorescence staining was performed on histologic sections of the pancreas of the different experimental groups. Mouse anti-cleaved caspase 3 antibody was used, and then, after that, Alexa 488-labeled goat anti-mouse (*n* = 3). Source data are available online for this figure.

ruled out because downstream *Kras^G12D* signaling was similar between cells expressing or not NUPR1. The second option was supported by several findings: (i) SGs are induced to protect the cells against several stresses, including the oncogenic stress; (ii) NUPR1 expression is activated in response to numerous stresses, including the oncogenic one; (iii) NUPR1 overexpression is sufficient to drive NUPR1-containing SGs formation; (iv) Inactivation of NUPR1 by ZZW-115 or siRNAs inhibits the NUPR1-driven SGs formation; and finally, (v) Inactivation of NUPR1 by ZZW-115 in *Pdx1-cre;LSL-Kras^G12D* mice results in blocking the PanINs development. If we assume that only cells expressing the mutated form of *Kras*, with an active NUPR1 stress protein, are capable of responding by activating the oncogenic stress response through SGs formation, we can then propose that the oncogenic stress promotes cell death rather than transformation in NUPR1-inactivated cells. An interesting point supporting this hypothesis is that in the pancreas from *Pdx1-cre;LSL-Kras^G12D* mice treated

with ZZW-115, only few cells expressing *Kras^G12D* were detected, however, massive levels of cleaved caspase 3 were found on these cells, presenting apoptotic features such as DNA fragmentation and apoptotic bodies. This can be explained by the fact that activation of *Kras* induced the oncogenic stress, and these cells, therefore, needed an efficient response to stress to survive. These in vivo results are further supported by the fact that, while cells having the *Kras^G12D*-activated are sensitive to NUPR1 inhibition, the counterpart cells with the inactivated *Kras^G12D* remain insensitive, demonstrating a strong dependency of those *Kras^G12D* cells on NUPR1 expression.

In a previous work Grabocka and Bar-Sagi reported that SGs are significantly elevated in mutant *Kras^G12D* cells after exposure to stress-inducing stimuli and that this upregulation is dependent on the production of the signaling lipid molecule 15-deoxy-delta 12,14 prostaglandin J2, conferring cyto-protection against stress stimuli and chemotherapeutic agents (Grabocka and Bar-Sagi 2016).

Taking into consideration the large number of stimuli (Santofimia-Castaño et al, 2019b) that induce NUPR1 expression and protein levels, the demonstration that this protein plays a critical role in the formation of an important population of SGs, suggests, therefore, that potential mediators of these phenomena are much larger than anticipated. The data we report here significantly extend the mechanism used by *Kras*$^{G12D}$ to form SGs by identifying the structural and functional contributions of NUPR1. This opens a new way towards novel therapeutics approaches to fight against the *Kras* oncogenic activity and in preventing cellular transformation in pancreas tissue. Mechanistically, we show that, in response to the oncogenic stress induced by mutated *Kras*, NUPR1-dependent SGs play a protective role against this stress, which would be essential for the transforming activity, since in their absence, cells would die instead of being transformed. Interestingly, short-term treatment of mice carrying PanINs with ZZW-115 blocked formation of the NUPR1-dependent SGs; but a longer treatment prevented the progression of PanINs towards PDAC, thus, indicating that targeting NUPR1-dependent SG is an excellent therapeutic strategy.

This new knowledge extends our understanding of pathophysiological mechanisms responsible for pancreatic cancer development. By using cells and animal models, we provide preclinical proof-of-concept suggesting that inhibition of NUPR1-dependent SGs formation could be utilized as a synthetic lethality therapeutic strategy in mutated *Kras*$^{G12D}$-dependent tumors.

# Methods

## Reagents and tools

See Table 1.

**Table 1.  Reagents and tools.**

| |
| --- |
| Imidazole, Sigma, Madrid, Spain |
| Trizma base and acid, Sigma, Madrid, Spain |
| DNase, Sigma, Madrid, Spain |
| SIGMAFAST protease tablets, Sigma, Madrid, Spain |
| NaCl, Sigma, Madrid, Spain |
| Ni2 + -resin, Sigma, Madrid, Spain |
| β-mercaptoethanol, BioRad, Madrid, Spain |
| Ampicillin, Apollo Scientific, Stockport, UK |
| Isopropyl-β-D-1-thiogalactopyranoside, Apollo Scientific, Stockport, UK |
| Triton X-100, VWR, Barcelona, Spain |
| TCEP, VWR, Barcelona, Spain |
| dialysis tubing with a molecular weight cut-off of 3500 Da, VWR, Barcelona, Spain |
| PAGEmark Tricolor, VWR, Barcelona, Spain |
| Poly(ADP-ribose) (PAR) polymer, Trevigen, Minneapolis, USA |
| Amicon centrifugal devices with a molecular weight cut-off of 3 kDa, Millipore, Barcelona, Spain |
| MiaPaCa-2 cells, American Type Culture Collection (ATCC), USA |
| Dulbecco's modified Eagle's medium (DMEM), Thermo Fisher Scientific, France, 61965-026 |
| Fetal bovine serum, Hyclone, Fisher Scientific, Loughborough, UK, SV30180.03 |
| Complete media RPMI, Thermo Fisher Scientific, France, 61870010 |
| Doxycycline, Sigma, France, 24390 |
| Amylase Activity Assay Kit, Sigma-Aldrich, France, MAK009 |
| 1-cm-pathlength quartz cell, Hellma, Kruibeke, Belgium |
| Anti-Flag M2-coated beads, Millipore Sigma, France, F3165 |
| GFP-Trap Agarose, Chromotek, GTA-10, Planegg-Martinsried, Germany |
| Flag peptide, Millipore Sigma, France, F3290 |
| NuPAGE 4–12% Bis-Tris acrylamide gels, Thermo Fisher Scientific, France |
| high-sequencing-grade trypsin, Promega, France |
| Sodium arsenate, Sigma, France, A6756 |
| PrestoBlue™ reagent, Life Technologies, Paris, France |
| Duolink In Situ, Merck, Darmstadt, Germany |
| Anti-NUPR1, rabbit, homemade |
| Anti-G3BP1, mouse, Abcam, Cambridge, UK, ab56574 |
| Anti-Flag, mouse, Millipore Sigma, France, F1804 |
| Goat anti-mouse Alexa Fluor 568, Thermo Fisher Scientific, France, A110314 |
| Donkey anti-rabbit Alexa Fluor 488, Thermo Fisher Scientific, France, A32790 |

**Table 1.** (continued)

| |
|---|
| Donkey anti-rabbit Alexa 647, Thermo Fisher Scientific, France, A31573 |
| DAPI, Thermo Fisher Scientific, France, D1306 |
| Mouse anti-G3BP1, Abcam, Cambridge, UK, ab56574 |
| Mouse anti-PABP, Abcam, Cambridge, UK, GR3398464 |
| Rabbit anti-p-EIF2α, Cell signaling Technology, Boston, US, 3398 |
| Rabbit anti-p-ERK1/2, Cell Signaling Technology, Boston, US, 4376 |
| Rabbit anti-p-AKT, Cell Signaling Technology, Boston, US, 9271 |
| Rabbit anti-KRASG12D, Genetex, Irvine, CA, US, GTX635362 |
| Rabbit anti-CK19, Abnova; Millipore, Molsheim, France, PAB12676 |
| Rabbit anti-amylase, Abcam, Cambridge, UK, ab21156 |
| Mouse anti-cleaved-caspase 3 (Asp175), Affinity Biosciences, Cincinnati, OH HQ, US, BF0711 |
| Alexa 568 goat anti-mouse, Thermo Fisher Scientific, France, A11031 |
| Alexa 488-donkey anti-rabbit, Thermo Fisher Scientific, France, A32790 |
| Anti-ERK1/2, Merck-Calbiochem, Darmstadt, Germany, M5670 |
| Anti-p-ERK1/2, Cell Signaling Technology, Boston, US, 4376 |
| Anti-AKT rabbit, Cell Signaling Technology, Boston, US, 3063 |
| Anti-p-AKT, rabbit, Cell Signaling Technology, Boston, US, 9271 |
| Anti-vinculin, Abcam, Cambridge, UK, ab129002 |
| Anti-β-actin, mouse, Sigma, France, A5316 |
| ECL detection system, Millipore Corp., Bedford, MA |
| Accutase, Thermo Fisher Scientific, France, A1110501 |
| Pacific-Blue annexin V, BioLegend, San Diego CA, US |
| Propidium iodide, Miltenyi Biotec, Bergisch Gladbach, Germany |
| IncucyteS3 Live-Cell Analysis System, Sartorius, Göttingen, Germany |
| CytoTox-ONE assays, Promega, France, G7890 |
| Caspase-Glo 3/7 assays, Promega, France, G8091 |
| Oligomycin, Millipore Sigma, Saint-Quentin-Fallavier, France |
| FCCP, Millipore Sigma, Saint-Quentin-Fallavier, France |
| Rotenone, Millipore Sigma, Saint-Quentin-Fallavier, France |
| Antimycine A, Millipore Sigma, Saint-Quentin-Fallavier, France |
| Go Script kit, Promega, France |
| INTERFERin reagent, Polyplus-transfection, Illkirch, France |
| siRNA control, Dharmacon, France, D-001810-10-05 |
| Mouse siRNA Nupr1, Dharmacon, France, L-049433-01-0010 |
| Mouse G3bp1 siRNA, Dharmacon, France, L-048735-01-0010 |
| Lipofectamine 3000 Transfection Reagent, Thermo Fisher Scientific, France, L3000015 |

## Materials

Imidazole, Trizma base and acid, DNase, SIGMAFAST protease tablets, NaCl and $Ni^{2+}$-resin were from Sigma (Madrid, Spain). The β-mercaptoethanol was from BioRad (Madrid, Spain). Ampicillin and isopropyl-β-D-1-thiogalactopyranoside were obtained from Apollo Scientific (Stockport, UK). Triton X-100, TCEP, dialysis tubing with a molecular weight cut-off of 3500 Da and the SDS protein marker (PAGEmark Tricolor) were from VWR (Barcelona, Spain). Poly(ADP-ribose) (PAR) polymer was from Trevigen (Minneapolis, USA). Amicon centrifugal devices with a molecular weight cut-off of 3 kDa were from Millipore (Barcelona, Spain). The rest of the used materials were of analytical grade. Water was deionized and purified on a Millipore system. Please see Appendix Table S1 for a complete list of reagents.

## Protein expression and purification

Wild-type NUPR1 and T68Q, A33Q, A33Q/T68Q NUPR1 mutants were produced and purified in LB media as described before (Santofimia-Castaño et al, 2022). The strain used for protein expression was NEB® Stable Competent *E. coli* (High Efficiency) C3040H. Protein concentrations were determined by ultraviolet (UV) absorbance, employing an extinction coefficient at 280 nm estimated from the number of tyrosines (in particular, NUPR1 has only two tyrosine residues) (Gill and von Hippel, 1989).

## Cell lines and cell culture

MiaPaCa-2 cells were obtained from the American Type Culture Collection (ATCC, Manassas, VA, USA) and cultured in

Dulbecco's modified Eagle's medium (DMEM, Thermo Fisher Scientific, 61965-026) containing 10% fetal bovine serum (Hyclone, SV30180.03) in an incubator with 5% $CO_2$ at 37 °C, and were authenticated by ATCC as a custom service. Primary mouse PDAC cells from *Nupr1^{wt} Pdx1-cre;LSL-Kras^{G12D};Ink4a/Arf^{fl/fl}* and *Nupr1^{ko};Pdx1-cre;LSL-Kras^{G12D};Ink4a/Arf^{fl}/fl were developed in our laboratory from male mice and* cultured in serum-free ductal media (SFDM) at 37°C in a 5% $CO_2$ incubator. i-*Kras* cell lines 4292, 9805, and 4668 were maintained in complete media RPMI (Thermo Fisher Scientific, 61870010) with 10% fetal bovine serum and 1 μg/mL doxycycline (Sigma, 24390) to continually express constitutively active *Kras*^{G12D} as described (Zhang et al, 2013), they were produced by Prof. Pasca di Magliano labs'. Primary pancreatic cancer cells were prepared and stored in aliquots at −140 °C until the time of their use and were cultured in serum free ductal media (SFDM). Cells have been regularly tested negative for mycoplasma.

## Mouse strains and tissue collection

*Pdx1-Cre;LSL-Kras*^{G12D} (KC) mice were obtained by crossing the following strains: *Pdx1-Cre* and *LSL-Kras*^{G12D} (Hingorani et al, 2003). Mice were kept within the Experimental Animal House of the Centre de Cancérologie de Marseille, pôle Luminy (Centre de Recherche en Cancérologie de Marseille). KC or WT mice were treated daily with 0.5% DMSO in physiologic serum (vehicle) or 5 mg/kg of ZZW-115 and killed after the treatment. Pancreases were fixed in 4% (wt/vol) formaldehyde. Male mice were housed under standard housing conditions with food and water ad libitum and all animal care and experimental procedures were performed in agreement with the Animal Ethics Committee of Marseille number 14 (C2EA-14). The criteria for inclusion were based on the requirement of accurate genotyping and the participants being male.

## Serum amylase activity

For determination of serum amylase activity, blood was collected after euthanasia of the mice by heart puncture. The blood was then centrifuged at $2000 \times g$ for 10 min at 4 °C to obtain serum. For amylase determination we used the Amylase Activity Assay Kit (MAK009, Sigma-Aldrich), following the protocol suggested by the manufacturer.

## Histological examination

Organs from mice were fixed in 4.0% formaldehyde, embedded in paraffin and then prepared into serial sections (4-μm thickness). The tissue sections were dewaxed and rehydrated and then were applied with hematoxylin-eosin (H&E) or haematoxylin-phloxine-saffron (HPS) staining. Images of the sections were taken by ZEISS Axio Imager Z2 microscope.

## Differential interference microscopy

Differential interference microscopy (DIC) was used to evaluate phase-separated liquid droplets of wild-type rNUPR1 or rNUPR1 mutated on positions A33Q, T68Q or A33Q/T68Q in presence of PEG-8000 5% and NaCl 50 mM pH 7.2 (Tris buffer) at 30 °C. Proteins were also incubated with PAR at 5 μM or RNA at 0.2 μg/μl

concentration. RNA was purified from MiaPaCa-2 cells. The wild-type rNUPR1 was also incubated with ZZW-115. Samples were added onto a fresh microscope slide (Fisher Scientific) which was immediately imaged on a Nikon Eclipse 90i microscope with 40X objective.

## Fluorescence

Fluorescence spectra were collected on a Cary Varian spectro-fluorometer (Agilent, Santa Clara, CA, USA), interfaced with a Peltier unit. All experiments were carried out at 25 °C. Following the standard protocols used in our laboratories, the samples were prepared the day before and left overnight at 5 °C; before experiments, samples were left for 1 h at 25 °C. A 1-cm-pathlength quartz cell (Hellma, Kruibeke, Belgium) was used. Fluorescence experiments were repeated in triplicates with newly prepared samples. Variations of results among the experiments were lower than 5%.

In the experiments aiming to determine the binding between wild-type NUPR1 and PAR, the concentrations of each macro-molecule were 4 μM. For experiments to detect the binding of RNA, protein and RNA concentrations were 100 ng/μL (12 μM of NUPR1 concentration). In all cases, the buffer used was 50 mM Tris (pH 7.5). Samples were excited at 280 nm with excitation and emission slits at 5 nm, with a data interval of 1 nm.

## Far-UV circular dichroism (CD)

Far-UV CD spectra were collected on a Jasco J810 spectropolarimeter (Jasco, Tokyo, Japan) with a thermostated cell holder and interfaced with a Peltier unit. The instrument was periodically calibrated with (+)-10-camphorsulfonic acid. A cell of pathlength 0.1 cm was used (Hellma, Kruibeke, Belgium). All spectra were corrected by subtracting the corresponding baseline. Concentration of each macromolecule and the buffers were the same used in the fluorescence experiments.

Isothermal wavelength spectra of each isolated macromolecule and those of the corresponding complex were acquired as an average of 6 scans, at a scan speed of 50 nm/min, with a response time of 2 s and a bandwidth of 1 nm. Samples were prepared the day before and left overnight at 5 °C to allow them to equilibrate. Before starting the experiments, samples were further left for 1 h at 25 °C.

## Flag- and GFP-NUPR1 co-immunoprecipitations and LC-MS/MS analysis

The experimental set-up was the same described previously (Lan et al, 2020). Briefly, MiaPaCa-2 cells, expressing Flag-NUPR1 or GFP-NUPR1 or their controls, were plated in 10 cm² dishes. When MiaPaCa-2 cells expressing Flag-NUPR1 or GFP-NUPR1 reached 70% confluence, they were treated for 24 h and lysed. Equal amounts of total protein were used to incubate with 30 μL of anti-Flag M2-coated beads (Millipore Sigma, F3165) or GFP-Trap Agarose (Chromotek, GTA-10). Beads were then washed 3 times, and proteins were eluted using ammonium hydrogen carbonate buffer containing 0.1 μg/μL of Flag peptide (Millipore Sigma, F3290). Eluted proteins were collected and loaded on NuPAGE 4–12% Bis-Tris acrylamide gels according to the manufacturer's

instructions (Invitrogen). Protein-containing bands were stained with Imperial Blue (Pierce), cut from the gel, and digested with high-sequencing-grade trypsin (Promega) before MS analysis. MS analysis was carried out by LC-MS/MS using an LTQ-Velos-Orbitrap or a Q Exactive Plus Hybrid Quadrupole-Orbitrap (Thermo Fisher Scientific) coupled online with a nanoLC Ultimate3000RSLC chromatography system (Dionex). Raw files generated from MS analysis were processed using Proteome Discoverer 1.4.1.14 (Thermo Fisher Scientific).

## Cell viability

Cells were plated in 96-well plates (5000 cells/well) overnight. Then, the media were supplemented with the compounds to be tested and the samples were incubated for another additional 72 h before performing the measurement. Cell viability was estimated after addition of PrestoBlue™ reagent (Life Technologies, Paris, France) for 3 h according to the protocol. Cell viability was normalized when comparing to untreated cell rates.

## Proximity ligation assay (PLA)

Fifty thousand MiaPaCa-2 cells were seeded on coverslips in 24-well plates. Twenty-four hours later, cells were treated. At the end of the experiment, cells were washed in phosphate buffer solution (PBS), fixed, and permeabilized. Immunostaining with Duolink In Situ (Merk), following the manufacturer's protocol, was done. Anti-NUPR1 (rabbit, homemade) and anti-G3BP1 (mouse, Abcam, ab56574) were used. Image acquisition was carried out in a confocal microscope, LSM 880 (x63 lens) controlled by Zeiss Zen Black. ImageJ (NIH) was used to count the number of green foci.

## Immunofluorescence of cultured cells

Cells were seeded in 24-well plates on coverslips and treated with ZZW-115 and Arsenate. After fixation, cells were incubated with the following primary antibodies at 1:100 dilution: mouse anti-G3BP1 (Abcam, ab56574), 1:100 Anti-NUPR1 (rabbit, homemade) or anti-Flag (mouse, Millipore Sigma, F1804). After washing steps, samples were incubated in the presence of secondary antibodies at 1:500 dilution (goat anti-mouse Alexa Fluor 568, A110314, donkey anti-rabbit Alexa Fluor 488, A32790 or donkey anti-rabbit Alexa 647, A31573 (all from Thermo Fisher Scientific). DAPI (D1306, Thermo Fisher Scientific) was used to stain the nucleus. Image acquisition of derived fluorescence and DAPI staining were performed by using an LSM 880 controlled by Zeiss Zen Black 63x lens. Analysis and measurement of the channels were conducted by using the ImageJ.

## Immunofluorescence staining of histology samples and tissue staining

Immunofluorescence staining was performed on 5-μm-thick paraffin-embedded tissue sections. The following antibodies were used at 1:100 dilution: mouse anti-G3BP1 (Abcam, ab56574), Anti-NUPR1 (rabbit, homemade), mouse anti-PABP (Abcam, GR3398464), rabbit anti-p-EIF2α (Cell signaling, 3398), rabbit anti-p-ERK1/2 (Cell Signaling, 4376), rabbit anti-p-AKT (Cell Signaling, 9271) rabbit anti-KRASG12D (Genetex GTX635362),

rabbit anti-CK19 (Abnova, PAB12676), rabbit anti-amylase (Abcam, ab21156), or mouse anti-cleaved-caspase 3 (Asp175) (Affinity Biosciences, BF0711). Alexa 568 goat anti-mouse (Thermo Fisher Scientific, A11031) and Alexa 488-donkey anti-rabbit (Thermo Fisher Scientific, A32790) secondary antibodies were used at 1:500 dilution. DAPI (D1306, Thermo Fisher Scientific) was used to stain the nucleus. Image acquisition of derived fluorescence and DAPI staining was performed using an LSM 880 controlled by Zeiss Zen Black 63x lens. Samples were also stained by Trichrome Stain Kit (ab150686) and with H&E and visualized on a microscope Zeiss Axio Imager 2. Percentage of affected area was calculated by ImageJ software.

## Western blotting

Proteins were resolved by SDS-PAGE and transferred to nitrocellulose membranes for 1 h. Then, membranes were blocked 1 h at room temperature with TBS (Tris-buffered saline solution) and 5% bovine serum albumin (BSA), and blotted overnight in TBS 5% BSA containing primary antibodies anti-KRAS$^{G12D}$ (1:1000, rabbit, Genetex GTX635362), anti-ERK1/2 (1:1000, Merc, M5670), anti-p-ERK1/2 (1:1000, rabbit, Cell Signaling, 4376), anti-AKT (1:1000, rabbit, Cell Signaling, 3063), anti-p-AKT (1:1000, rabbit, Cell Signaling, 9271), anti-G3BP1 (Abcam, ab56574), 1:100 anti-NUPR1 (rabbit, homemade), anti-vinculin (Abcam, ab129002) and anti-β-actin (1:1000, mouse, Sigma, A5316). After extensive washes in TBS 0.1% Tween-20, membranes were incubated 1 h at room temperature with HRP-conjugated secondary antibodies at 1:5000. Antigen/antibody complex was revealed using ECL detection system (Millipore) and visualized using a Pxi imaging device (SynGene).

## Cell viability

Cells were plated in 96-well plates (5000 cells/well). Twenty-four hours later, the media were supplemented with various concentrations of ZZW-115, added with a D300e Digital Dispenser (Tecan), in the presence or absence of doxycycline. Cell viability was estimated after 24 h of incubation by the addition of the PrestoBlue viability reagent (Thermo Fisher Scientific, A13261) for 3 h according to the protocol provided by the supplier. Cell viability was normalized with respect to untreated cell rates.

## Annexin V/PI staining

Cells were collected after incubation for 24 h of ZZW-115-treatment. Cells were washed and then detached with Accutase (Thermofisher, A1110501), and resuspended in annexin-binding buffer (home-made). Pacific-Blue annexin V (5 μL, BioLegend) was added to the cell suspension and incubated for 15 min. Before analysis by flow cytometry, propidium iodide (5 μL, Miltenyi Biotec) was added to the suspension. A MACSQuant-VYB (Miltenyi Biotec) was used to collect 10,000 events per sample. Data analysis was performed using FlowJo 10.7.1 software.

## Cell index measurement

Cells were seeded at a density of 5000 cells/well in 96-well plates. Cells were allowed to attach overnight and treated the next day for

30 h with increasing concentrations of ZZW-115 in the presence or absence of doxycycline. Cell count was measured by IncucyteS3 Live-Cell Analysis System (Sartorius), placed in an incubator maintained at 37 °C in a humidified 5% $CO_2$ atmosphere. Data of each well was normalized to the cell index at time 0 h, measured just before the treatment was added to the medium with a D300e Digital Dispenser (Tecan).

## LDH assay and caspase 3/7 activity assay

Cells were seeded at a density of 10,000 cells/well in 96-well plates. Cells were allowed to attach overnight and treated the next day for 24 h with different concentrations of ZZW-115 in the presence or absence of doxycycline. At the end of the experiment, LDH release and caspase 3/7 activity were monitored using CytoTox-ONE (Promega G7890) and Caspase-Glo 3/7 (Promega G8091) assays, respectively. Data were normalized with respect to the cell number.

## Measurement of mitochondrial oxidative phosphorylation (OXPHOS)

Fifty thousand cells/well were plated at 24-well plate (Seahorse) and incubated overnight. Cells were treated with doxycycline or MARTX1133 30 nM for 24 h. The Oxygen Consumption Rate (OCR) (pmol $O_2$/min) was measured using the Seahorse Bioscience XF24 Extracellular Flux Analyzer (Agilent) in response to 1 µM oligomycin, 0.25 µM carbonylcyanide p-(trifluoro-methoxy) phenylhydrazone (FCCP), 0.5 µM rotenone/Antimycine A (Millipore Sigma).

## qRT-PCR

Total RNA was extracted from cells using Trizol kit (Invitrogen) and cDNA was obtained by reverse-transcribed using Go Script kit (Promega), according to the manufacturer's instructions. Real-time quantitative PCR (qRT-PCR) was performed using AriaMx system (Agilent). Primer sequences are listed below: Nupr1-F: 5′-CC AATACCAACCGCCCTAGC-3′; Nupr1-R: 5′-CTGTGGTCTGG CCTTATCTCC-3′; G3bp1-F: ACCCCGTCATTCAGAGTTGC, G3bp1-R: TCCTCGTTGGAGTGACATCG Ctgf-F: CAGCTGG-GAGAACTGTGTACG; Ctgf-R: GTACACCGACCCACCGAAGA.

## siRNA and plasmids transfection

For siRNA transfection, cells were plated at 70% confluence and INTERFERin reagent (Polyplus-transfection) was used to perform siRNA transfections, according to the manufacturer's protocol. Scrambled siRNA that targets no known gene sequence was used as a negative control. All assays were carried out 48 h post-transfection. The sequence of NUPR1-specific siRNA was siNUPR1-1 r(GGAGGACCCAGGACAGGAU)dTdT. Commercial siRNAs were used for the transfection of the murine cells: siRNA control, (Dharmacon, D-001810-10-05), Mouse siRNA Nupr1 (Dharmacon, L-049433-01-0010) or mouse G3bp1 siRNA (Dharmacon, L-048735-01-0010). For plasmid transfection, 15.000 9805-i-*Kras* cells were seeded on coverslips in 12 wells plates and transfected with 1 µg of plasmid DNA (Nupr1-Flag, Nupr1-A33Q/T68Q-Flag, or G3BP1-GFP (Addgene: Plasmid #119950) using

**The paper explained**

**Problem**

Pancreatic ductal adenocarcinoma (PDAC), a highly lethal cancer, is initiated by a mutation in the *Kras* oncogene in over 90% of cases. This oncogenic activation creates a stressful situation for the cell, where the process of transformation requires the activation of an integrated stress response, coordinating cellular adaptation to stress and allowing for transformation. Among these factors, the intrinsically disordered protein (IDP) NUPR1, crucial for cellular adaptation to stress, plays significant roles in PDAC development, although its mechanism of action remains unclear. Given that NUPR1 is an IDP and that liquid–liquid phase separation (LLPS) is often promoted by IDPs, we hypothesized that the expression of NUPR1 could be involved in the formation of membrane-less organelles stress granules (SGs), which seem to be associated with *Kras*-dependent transformation.

**Results**

In this study, we demonstrate that NUPR1 expression is strongly activated by mutated *Kras* and subsequently drives LLPS for SGs development. Pharmacological inhibition of NUPR1 with ZZW-115 is sufficient to block LLPS and SGs formation in vitro. Additionally, we show that inhibiting NUPR1-dependent SGs formation with ZZW-115 in *Pdx1-cre;LSL-Kras^{G12D}* (KC) mice blocks the transformation process by inducing caspase 3 activation, DNA fragmentation, and the formation of apoptotic bodies, leading to cell death specifically in *Kras^{G12D}*-expressing cells. Moreover, short-term treatment of these mice with ZZW-115 blocks the formation of SGs, while longer treatment reverses or significantly retards the development of PanINs and, consequently, PDAC development.

**Impact**

This study proposes that targeting NUPR1-dependent SGs formation by using NUPR1 inhibitors could be a therapeutic approach to induce synthetic lethality in *Kras*-mutated-driven tumors.

Lipofectamine 3000 Transfection Reagent (Thermo Fisher Scientific, L3000015).

## Statistics

Statistical analyses were conducted by using the unpaired two-tailed Student t-test, or one-way ANOVA with Sidak correction or two-way ANOVA with Sidak correction. The results were expressed as the mean ± SD of at least three independent experiments. A *p*-value of <0.05 was regarded as statistically significant. The initial sample size for general inclusion in our study was set at $n = 3$, and for animal studies, it was selectively increased by 5 to 8 samples in specific cases. This approach aimed to increase statistical power, ensuring a 90% probability of detecting statistically significant deviations at $P < 0.05$. For animal experiments, mice were randomized in an unbiased fashion. Researchers were not blinded during mouse experiments.

# Data availability

This study includes no data deposited in external repositories.

# Peer review information

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

## Acknowledgements

We are grateful to Dr Arkaitz Carracedo for the critical comments on the manuscript. This work was supported by La Ligue Contre le Cancer [équipe labellisée 2022] to JI, INCa 2020-098 to JI, Fondation ARC to PSC, Comunidad Valenciana [CIAICO 2021/0135 to JLN] and INSERM to JI. MM is supported by La Ligue Contre le Cancer [équipe labellisée, EL2018.LNCC/MaM]. ME is supported by Fondation ARC. GL and RU were funded by NIH (R01DK052913), and they are grateful to AHW and the Linda and John Mellowes Center for their support. XL is recipient of the predoctoral fellowship from China Scholarship Council (CSC). We thank the imaging platform IBDM, Aix Marseille University, Marseille, France.

## Author contributions

**Patricia Santofimia-Castaño**: Conceptualization; Data curation; Formal analysis; Validation; Methodology; Writing—original draft; Writing—review and editing. **Nicolas Fraunhoffer**: Resources; Data curation; Formal analysis. **Xi Liu**: Data curation; Visualization; Methodology. **Ivan Fernandez Bessone**: Resources; Data curation; Methodology. **Marina Pasca di Magliano**: Resources. **Stephane Audebert**: Formal analysis; Methodology. **Luc Camoin**: Resources; Data curation; Formal analysis; Methodology. **Matias Estaras**: Resources; Data curation; Formal analysis; Validation; Investigation; Methodology. **Manon Brenière**: Methodology. **Mauro Modesti**: Conceptualization; Resources; Data curation; Validation; Investigation; Methodology; Writing—original draft. **Gwen Lomberk**: Resources; Data curation; Validation; Methodology; Writing—original draft. **Raul Urrutia**: Resources; Formal analysis; Supervision; Investigation; Methodology; Writing—original draft. **Philippe Soubeyran**: Conceptualization; Resources; Data curation; Formal analysis; Supervision; Investigation; Visualization; Methodology; Writing—original draft; Writing—review and editing. **Jose Luis Neira**: Conceptualization; Data curation; Formal analysis; Supervision; Investigation; Methodology; Writing—original draft; Writing—review and editing. **Juan Iovanna**: Conceptualization; Resources; Data curation; Formal analysis; Supervision; Funding acquisition; Validation; Investigation; Visualization; Methodology; Writing—original draft; Project administration; Writing—review and editing.

## Disclosure and competing interests statement

JI is co-founder of PanCa Therapeutics and PredictingMed. The other authors declare no competing interests. The funders had no role in the design of the study; in the collection, analyses, or interpretation of data, in the writing of the manuscript, or in the decision to publish the results.

# Expanded View Figures

**Figure EV1. NUPR1 is a key protein for the formation of SGs.**

(**A**) NUPR1 and G3BP1 mRNA levels were measured in 4292 i-*Kras*, 4668 i-*Kras* and 9805 i-*Kras* cells 48 h after transfection, expressed as fold changes ($n = 3$ independent experiments, triplicates were made on each one). Data represent mean ± SD. One-way ANOVA, Sidak correction. (**B**) Western-blot analysis was performed in MiaPaCa-2 cells to evaluate G3BP1, NUPR1 and vinculin levels ($n = 3$). (**C**) Immunofluorescence was performed in MiaPaCa-2 cells transfected with siControl or siNUPR1 and after 24 h, with G3BP1 or GFP plasmid, cells were fix 24 h later. Mouse anti-G3BP1 and Alexa 568-labeled goat anti-mouse secondary antibodies were used. A representative experiment is shown ($n = 2$). (**D**) Cell count of MiaPaCa-2 cells in the previous conditions was evaluated ($n = 3$ independent experiments, triplicates were made on each one), data represent mean ± SD, two-way ANOVA, Sidak correction. (**E**) Immunofluorescence was performed in PDAC primary cell lines, mouse anti-G3BP1 and Alexa 568-labeled goat anti-mouse secondary antibodies were used ($n = 3$). (**F**) Chemogram assays were done on pancreatic cancer cell lines with increasing concentrations of ZZW-115 for 72 h ($n = 3$). Source data are available online for this figure.

▶

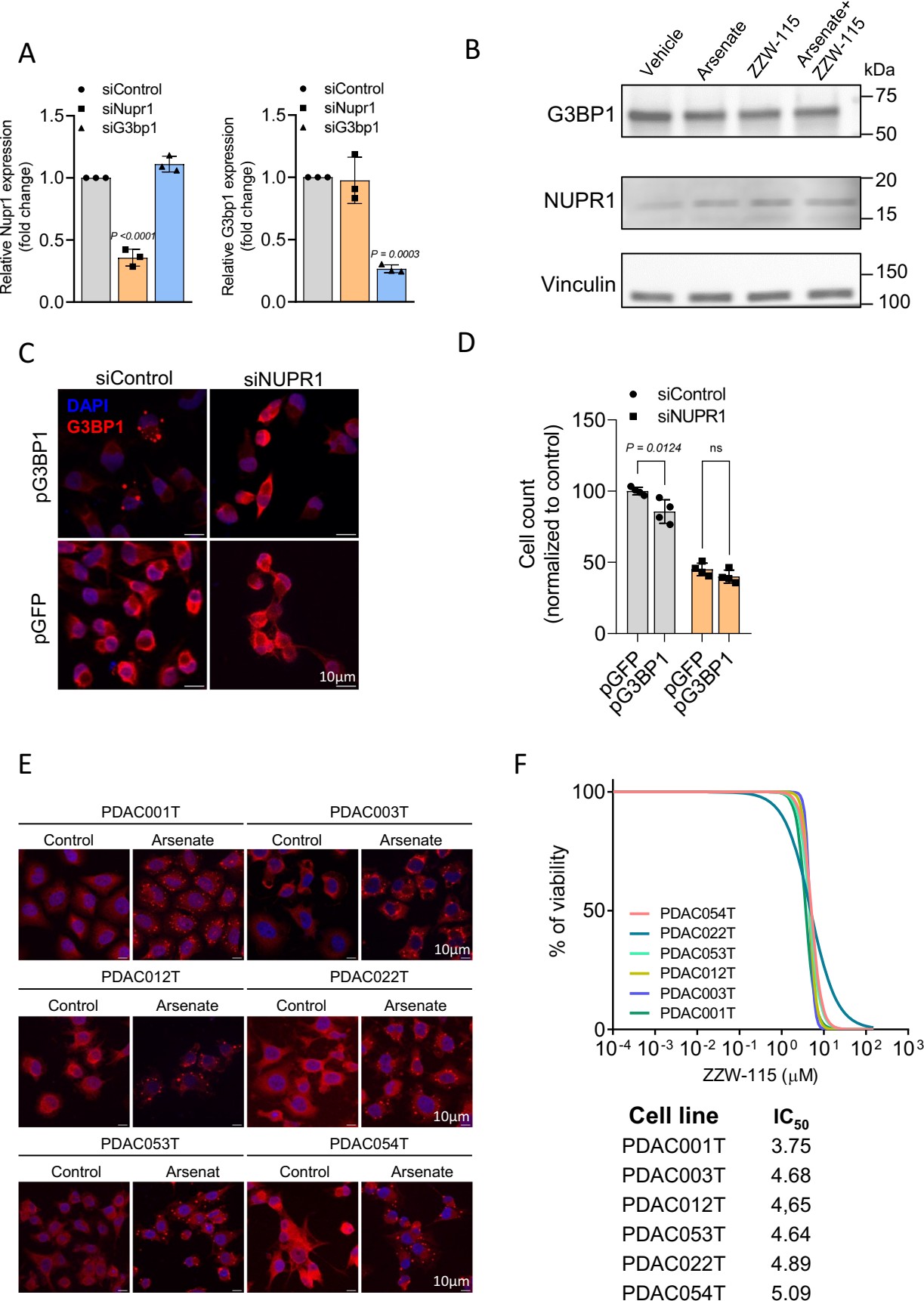

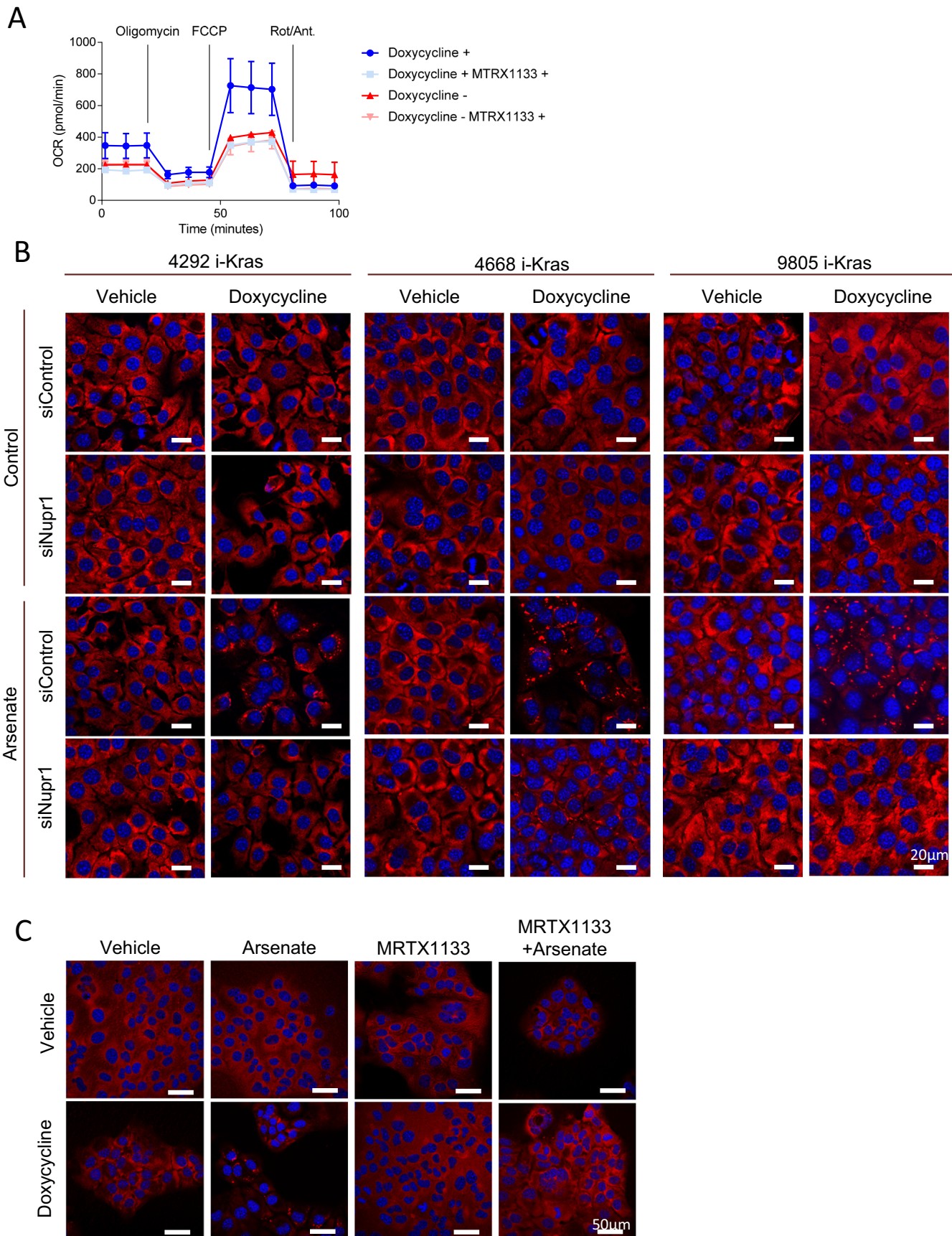

◄ **Figure EV2.   Inhibition of NUPR1 by siRNA or *Kras* inhibitor MRTX1133 prevents SGs formation in i-*Kras* cells.**

(**A**) OXPHOS metabolism, reflected by oxygen consumption rate (OCR) levels, were measured in 9805 i-*Kras* cells in the absence of presence of doxycycline and/or 30 nM of MRTX1133 for 24 h, a representative experiment is shown (data represent mean ± SEM, $n = 3$ independent experiments, triplicates were made on each one). (**B**) Immunofluorescence staining was performed in 4292 i-*Kras*, 4668 i-*Kras* and 9805 i-*Kras* cells 48 h after transfection with siControl of siNUPR1. Mouse anti-G3BP1 and then, Alexa 568-labeled goat anti-mouse secondary antibody were used ($n = 3$). (**C**) Immunofluorescence staining was performed in 9805 i-*Kras* cells in the absence of presence of arsenate and/or 30 nM of MRTX1133 for 24 h. Mouse anti-G3BP1 and Alexa 568-labeled goat anti-mouse secondary antibodies were used ($n = 3$). Source data are available online for this figure.

A

**Control**
number of G3BP1/Flag
co-localizing dots by nucleus

● Doxycycline –
◇ Doxycycline +

*P = 0.0003  P = 0.0001*
*P = 0.0003  P = 0.0001*

NUPR1-Flag  NUPR1^A33Q/T68Q-Flag  GFP

**Arsenate**
number of G3BP1/Flag
co-localizing dots by nucleus

● Doxycycline –
◇ Doxycycline +

*P <0.0001*
*P <0.0001*  *P = 0.0005*
*P <0.0001*

NUPR1-Flag  NUPR1^A33Q/T68Q-Flag  GFP

**Arsenate+ZZW-115**
number of G3BP1/Flag
co-localizing dots by nucleus

● Doxycycline –
◇ Doxycycline +

*P <0.0001  P <0.0001*
*P <0.0001  P <0.0001*
*P <0.0001*

NUPR1-Flag  NUPR1^A33Q/T68Q-Flag  GFP

B

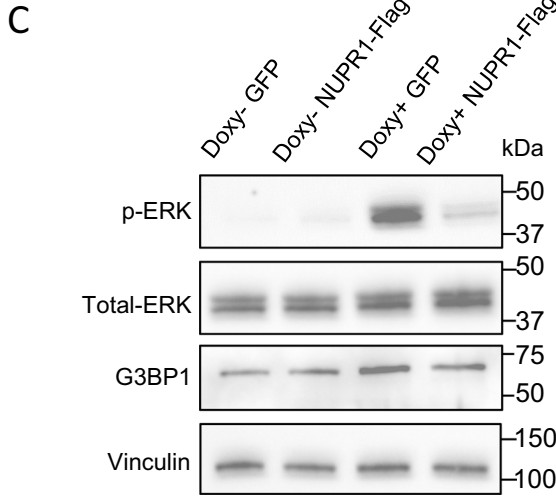

|  | NUPR1-Flag | | NUPR1^A33Q/T68Q-Flag | | GFP | |
|---|---|---|---|---|---|---|
|  | Vehicle | Doxycycline | Vehicle | Doxycycline | Vehicle | Doxycycline |

ZZW-115 — G3BP1 / Flag / Merge

20μm

● Doxycycline –
◇ Doxycycline +

number of SGs by nucleus

*P = 0.0001  P = 0.0001*
*P <0.0001  P <0.0001*

NUPR1-Flag  NUPR1^A33Q/T68Q-Flag  GFP

C

Doxy- GFP  Doxy- NUPR1-Flag  Doxy+ GFP  Doxy+ NUPR1-Flag  kDa

p-ERK                                    —50
                                         —37

Total-ERK                                —50
                                         —37

G3BP1                                    —75
                                         —50

Vinculin                                 —150
                                         —100

◀ **Figure EV3. ZZW-115 prevents SGs formation in cells overexpressing NUPR1 independent of *Kras* signaling.**

(A) Quantification of number of G3BP1 or NUPR1 by nucleus or number of colocalizing NUPR1/G3BP1 dots by nucleus in 9805 i-*Kras* cells is shown ($n = 5$). Data represent mean ± SD, two-way ANOVA with Sidak correction. (B) Immunofluorescence staining was performed in 9805 i-*Kras* cells 24 h post-transfection of NUPR1-Flag wild-type, its double mutant NUPR1 A33Q /T68Q-Flag, or a GFP plasmids upon treatment with ZZW-115 at 6 µM for 6 h. Mouse anti-G3BP1 and rabbit anti-Flag and then, Alexa 568-labeled goat anti-mouse and Alexa 647-labeled goat anti-rabbit secondary antibodies were used ($n = 5$ independent experiments, 5 pictures were used to calculate the mean of each experiment). Data represent mean ± SD, two-way ANOVA with Sidak correction. (C) Western blot analysis was performed in 9805 i-*Kras* cells to evaluate p-ERK, total-ERK, G3BP1 and vinculin levels ($n = 3$). Source data are available online for this figure.

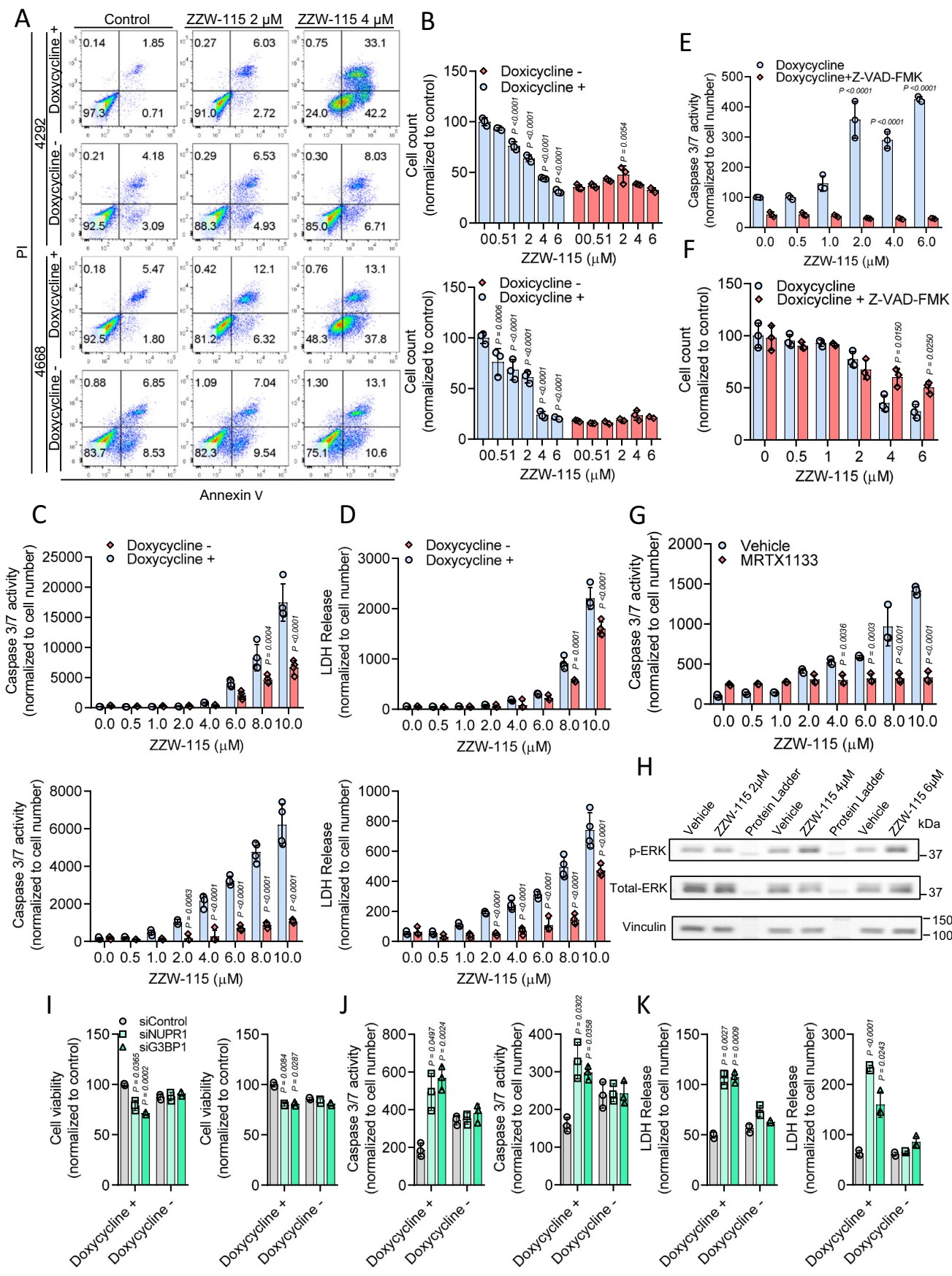

**Figure EV4.   Inhibition of NUPR1 induced cell death in *Kras^G12D*-activated cells.**

(A) Flow cytometry analysis of annexin V/PI staining of 4292 i-*Kras* (up), 4668 i-*Kras* (down) cells, following 24 h of treatment with increasing concentrations of ZZW-115 in presence or in the absence of doxycycline was done. A representative experiment of the dot plot profile of cells is shown ($n = 3$). (B) Cell count of 4292 i-*Kras* (up), 4668 i-*Kras* (down) cells measured by IncuCyte live-cell imaging after 30 h of treatment of increasing concentrations of ZZW-115 in the presence or in the absence of doxycycline was evaluated ($n = 3$ independent experiments, triplicates were made on each one). Data represent mean ± SD, Two-way ANOVA with Sidak correction. 4292 i-*Kras* (up), 4668 i-*Kras* (down) cells were incubated at increasing concentrations ZZW-115 in presence or in the absence of doxycycline for 24 h and (C) caspase 3/7 activity ($n = 3$) and (D) LDH release were measured ($n = 3$). For both, data represent mean ± SD, Two-way ANOVA with Sidak correction. (E) Caspase 3/7 activity was measured in 9805 i-*Kras* cells after 24 h of treatment of increasing concentrations of ZZW-115 in the presence or in the absence of Z-VAD-FMK. Data represent mean ± SD, ($n = 3$) two-way ANOVA with Sidak correction. (F) Cell count was measured in 9805 i-*Kras* cells after 24 h of treatment of increasing concentrations of ZZW-115 in the presence or in the absence of Z-VAD-FMK ($n = 3$). Data represent mean ± SD, two-way ANOVA with Sidak correction. (G) Caspase 3/7 activity was measured in 9805 i-*Kras* cells after 24 h of treatment at increasing concentrations of ZZW-115 in the presence or in the absence of 30 nM of MRTX1133 ($n = 3$). Data represent mean ± SD, two-way ANOVA with Sidak correction. (H) Western blot analysis was performed in 9805 i-*Kras* cells to evaluate p-ERK, total-ERK and vinculin levels upon ZZW-115-treatment ($n = 3$). (I) Cell viability (J) caspase 3/7 activity and (K) LDH release were measured in 4292 i-*Kras* (left), 4668 i-*Kras* (right) cells transfected with siControl, siG3BP1 or siNupr1 for 48 h in the presence or absence of doxycycline ($n = 3$ independent experiments, triplicates were made on each one). Data represent mean ± SD, Two-way ANOVA with Sidak correction. Source data are available online for this figure.

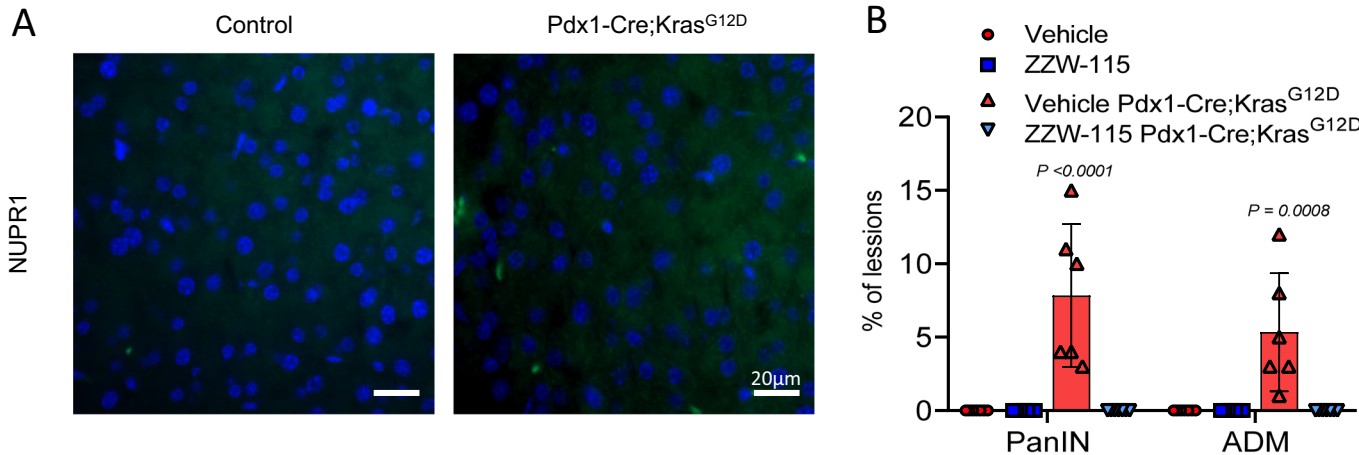

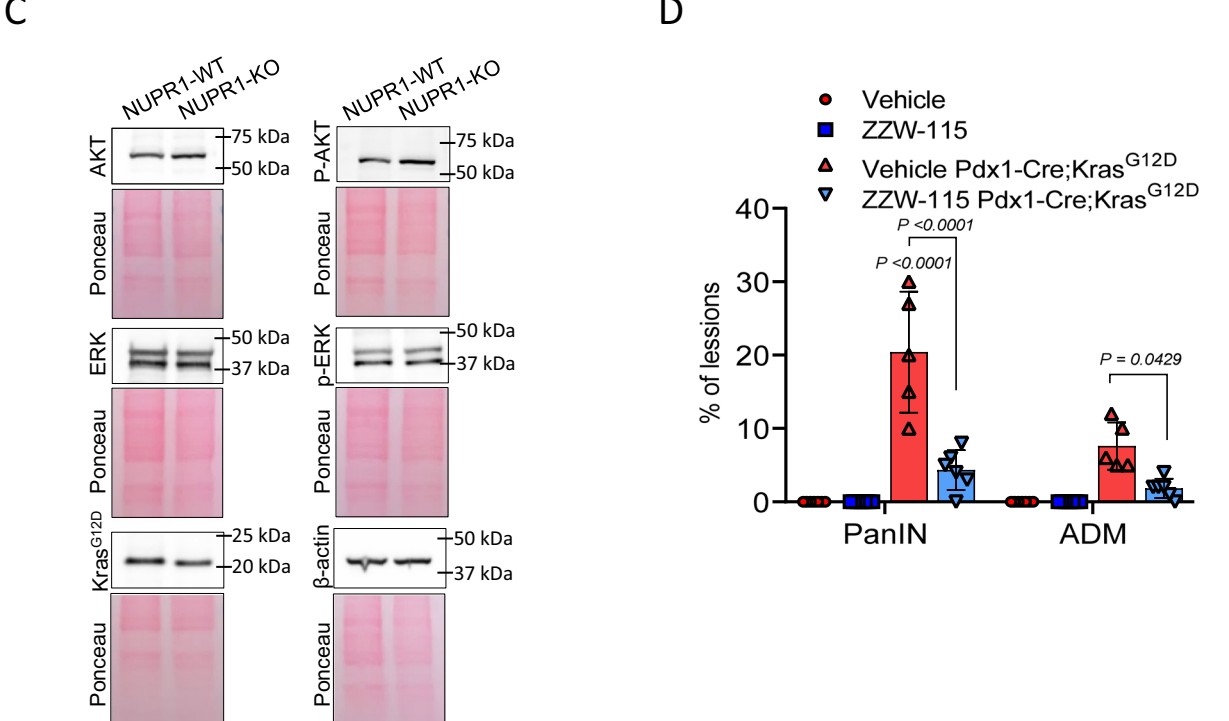

**Figure EV5. NUPR1 inhibition induced cell death in vivo.**

(A) Immunohistofluorescence staining was performed on histologic sections of the pancreas of the different experimental groups at 5 weeks of age. Rabbit anti-NUPR1 antibody was used, then, Alexa 488-labeled goat anti-rabbit ($n = 3$). (B) Percentage of tissue affected by ADM and PanIN lesions per tissue field in Control and Pdx1-Cre;Kras$^{G12D}$ mice treated with vehicle solution or ZZW-115 for ten weeks (from 5 to 15 weeks) ($n = 6$). Data represent mean ± SD, Two-way ANOVA with Sidak correction. (C) Western-blot analysis was performed in *Pdx1-cre;LSL-Kras$^{G12D}$/INK4a/Arf$^{fl/fl}$/NUPR1$^{+/+}$* and *Pdx1-cre;LSL-Kras$^{G12D}$/INK4a/Arf$^{fl/fl}$/NUPR1$^{-/-}$* mice cells to evaluate AKT, p-AKT, ERK, p-ERK, *Kras$^{G12D}$* or β-actin levels. (D) Percentage of tissue affected by ADM and PanIN lesions per tissue field in Control and Pdx1-Cre;Kras$^{G12D}$ mice treated with vehicle solution or ZZW-115 for four weeks (from 15 to 19 weeks) ($n = 5$). Data represent mean ± SD, Two-way ANOVA with Sidak correction. Source data are available online for this figure.

