## [Peer Review File · EMBO Molecular Medicine]

Targeting NUPR1-Dependent Stress Granules Formation to Induce Synthetic Lethality

Juan Iovanna, Patricia Santofimia-Castaño, Nicolas Fraunhoffer, Xi Liu, Ivan Fernandez Bessone¹, Marina Pasca di Magliano, Stephane Audebert, Luc Camoin, Matias Estaras, Manon Brenière, Mauro Modesti, Gwen Lomberk, Raul Urrutia, Philippe Soubeyran, and Jose Neira

DOI: [10.15252/emmm.202318852](https://doi.org/10.15252/emmm.202318852)

Corresponding authors: Juan Iovanna (juan.iovanna@inserm.fr), Patricia Santofimia-Castaño (patricia.santofimia@inserm.fr)

Review Timeline:

Submission Date:	18th Oct 23
Editorial Decision:	16th Nov 23
Revision Received:	30th Dec 23
Editorial Decision:	9th Jan 24
Revision Received:	16th Jan 24
Editorial Decision:	19th Jan 24
Revision Received:	22nd Jan 24
Accepted:	25th Jan 24

Editor: Lise Roth

Transaction Report:

16th Nov 2023

Dear Dr. Iovanna,

Thank you for the submission of your manuscript to EMBO Molecular Medicine. We have now received feedback from the three reviewers who agreed to evaluate your manuscript. As you will see from the reports below, the referees acknowledge the interest of the study and are overall supporting publication of your work pending appropriate revisions.

Addressing the reviewers' concerns in full will be necessary for further considering the manuscript in our journal, and acceptance of the manuscript will entail a second round of review. EMBO Molecular Medicine encourages a single round of revision only and therefore, acceptance or rejection of the manuscript will depend on the completeness of your responses included in the next, final version of the manuscript. For this reason, and to save you from any frustrations in the end, I would strongly advise against returning an incomplete revision.

We are expecting your revised manuscript within three months, if you anticipate any delay, please contact us.

We require:

4) A .docx formatted letter INCLUDING the reviewers' reports and your detailed point-by-point responses to their comments. As part of the EMBO Press transparent editorial process, the point-by-point response is part of the Review Process File (RPF), which will be published alongside your paper.

5) A complete author checklist, which you can download from our author guidelines (<https://www.embopress.org/page/journal/17574684/authorguide#submissionofrevisions>). Please insert information in the checklist that is also reflected in the manuscript. The completed author checklist will also be part of the RPF.

6) Please note that all corresponding authors are required to supply an ORCID ID for their name upon submission of a revised manuscript. An ORCID identifier is currently missing for Dr. Patricia Santofimia-Castaño.

7) It is mandatory to include a 'Data Availability' section after the Materials and Methods. Before submitting your revision, primary datasets produced in this study need to be deposited in an appropriate public database, and the accession numbers and database listed under 'Data Availability'. Please remember to provide a reviewer password if the datasets are not yet public (see <https://www.embopress.org/page/journal/17574684/authorguide#dataavailability>).

8) For data quantification: please specify the name of the statistical test used to generate error bars and P values, the number (n) of independent experiments (specify technical or biological replicates) underlying each data point and the test used to calculate p-values in each figure legend. The figure legends should contain a basic description of n, P and the test applied. Graphs must include a description of the bars and the error bars (s.d., s.e.m.). Please provide exact p values.

9) Our journal encourages inclusion of *data citations in the reference list* to directly cite datasets that were re-used and obtained from public databases. Data citations in the article text are distinct from normal bibliographical citations and should directly link to the database records from which the data can be accessed. In the main text, data citations are formatted as

follows: "Data ref: Smith et al, 2001" or "Data ref: NCBI Sequence Read Archive PRJNA342805, 2017". In the Reference list, data citations must be labeled with "[DATASET]". A data reference must provide the database name, accession number/identifiers and a resolvable link to the landing page from which the data can be accessed at the end of the reference. Further instructions are available at .

11) For more information: There is space at the end of each article to list relevant web links for further consultation by our readers. Could you identify some relevant ones and provide such information as well? Some examples are patient associations, relevant databases, OMIM/proteins/genes links, author's websites, etc...

12) Author contributions: CRediT has replaced the traditional author contributions section because it offers a systematic machine readable author contributions format that allows for more effective research assessment. Please remove the Authors Contributions from the manuscript and use the free text boxes beneath each contributing author's name in our system to add specific details on the author's contribution. More information is available in our guide to authors.

13) Disclosure statement and competing interests: We updated our journal's competing interests policy in January 2022 and request authors to consider both actual and perceived competing interests. Please review the policy <https://www.embopress.org/competing-interests> and update your competing interests if necessary.

14) Every published paper now includes a 'Synopsis' to further enhance discoverability. Synopses are displayed on the journal webpage and are freely accessible to all readers. They include a short stand first (maximum of 300 characters, including space) as well as 2-5 one-sentences bullet points that summarizes the paper. Please write the bullet points to summarize the key NEW findings. They should be designed to be complementary to the abstract - i.e. not repeat the same text. We encourage inclusion of key acronyms and quantitative information (maximum of 30 words / bullet point). Please use the passive voice. Please attach these in a separate file or send them by email, we will incorporate them accordingly.

15) As part of the EMBO Publications transparent editorial process initiative (see our Editorial at <http://embomolmed.embopress.org/content/2/9/329>), EMBO Molecular Medicine will publish online a Review Process File (RPF) to accompany accepted manuscripts.

In the event of acceptance, this file will be published in conjunction with your paper and will include the anonymous referee reports, your point-by-point response and all pertinent correspondence relating to the manuscript. Let us know whether you agree with the publication of the RPF and as here, if you want to remove or not any figures from it prior to publication. Please note that the Authors checklist will be published at the end of the RPF.

I look forward to receiving your revised manuscript.

Yours sincerely,

Lise Roth

**** Reviewer's comments ****

Referee #1 (Comments on Novelty/Model System for Author):

No suggestions to improve

Referee #1 (Remarks for Author):

In this manuscript Santofimia-Castaño et al demonstrate that the transcriptional regulators NUPR1 mediates the formation of the non-membranous organelles known as stress granules (SGs) in pancreatic cancer cells and pancreatic intraepithelial neoplasia (PanINs). The authors nicely demonstrate that NUPR1-dependent SG formation is mediated by the capacity of NUPR1 to undergo liquid-liquid phase separation (LLPS) via its intrinsically disordered region. A synthetic compound ZZW-115 that binds to and blocks residues critical to LLPS blocks SG formation. Pancreatic cancer is a very aggressive and therapy-resistant disease for which few targeted therapies exist. Enhanced SG formation by mutant KRAS has been shown to contribute to the resistance of mutant KRAS cancer cells to cell stress, and more recently SGs were shown to play an important role in pancreatic tumorigenesis (PMID: 35674408). As such, SG biology and their targeting has important implications for pancreatic cancer therapeutics. The authors show that mutant KRAS upregulates NUPR1 expression and that genetic or pharmacological inhibition of NUPR1 impedes SG formation in vitro and in mouse models of PanINs. Notably, they show that ZZW-115 blocks SG formation and PanIN development and expansion. The findings of this study are novel, timely, and have important therapeutic implications. A couple of minor comments are listed below:

1. A schematic of the domains/structure of NUPR1 and the positioning of the IDR region and the A33Q, T68Q or A33Q/T68Q residues that are targeted by ZZW-115 would help to visualize/make clear the relevance of these residues to LLPS and how ZZW-115 may impair LLPS.
2. The data that NUPR1 inhibition or knockdown leads to SG inhibition, cell death, and impedes PanIN development and growth are very clear. However, NUPR1 has additional interactions/functions than the demonstrated binding to SG core proteins under stress and promoting SG formation. The authors cannot rule out that the observed effects are not due to such additional functions. This can be tested by rescuing SG formation through using synthetic constructs or hyperactive G3bp1 mutants (PMID: 32302571; PMID: 35674408). While the lack of these experiments does not detract from the novelty and significance of the findings, it should be discussed in the manuscript in the discussion section.

Referee #2 (Remarks for Author):

The authors observed that the stress-inducible nuclear protein NUPR1 is involved in the development of stress granules (SG). Under oncogenic stress, mediated by the expression of mutated KRAS, NUPR1 is essential for SG granule formation and the prevention of KRAS-driven caspase-mediated apoptosis. Consequently, treatment with the NUPR1 inhibitor ZZW-115, prevented/reduced PanIN formation and progression. This is a very important contribution opening the avenue for the chemoprevention of a deadly disease.

Considerations:

- 1) The effects of ZZW-115 on early ADM and PanIN development are impressive (Fig.5) as are the effects on later stages (Fig. 9). Are treatment data with ZZW-115 also available for more aggressive models or is the ZZW-115 effect dependent on tumor suppressors like CDKN2A or TP53? The authors treated the animals over longer periods with ZZW-115. Toxicity of the compound? Toxicity monitored? Was the on-target activity measured?
 - 2) The authors argue that "inhibition of NUPR1-dependent SGs could be utilized as a synthetic lethality therapeutic strategy in mutated KrasG12D-dependent tumors". Considering the heterogeneity of the human disease, it would be important to provide some insight into a larger cohort of cellular models on how common and heterogeneous SG formation is in established human and murine pancreatic cancer cell lines. Is the number of stress granules correlating with the therapeutic efficacy of ZZW-115?
- Minor points:

"Among these sensors, the serine/threonine kinases promote phosphorylation of eIF2 α , a main inductor of stress granules (SGs) formation, a membrane-less organelle (MLO) sensed to protect the cell against the stress (McInerney et al, 2005)." There are several current reviews on SGs - a very detailed and critical is written by Glauninger et al. in Molecular Cell. Worth to cite? Figure 2C: a NUPR1 knockdown control by western blot is needed. It would also be important to see protein expression of NUPR1 and G3BP1 by western blot, to control the arsenite-dependent induction. It is also unclear, how SGs are visualized in 2C. The y-axis is labeled as SGs per nucleus, which might be a vague description for the cytoplasmic SGs. Perhaps it is easier to follow the manuscript if the panels are in the reading order.

Mycoplasma testing protocol not reported.

Figure 3: Doxycycline concentration too high? Might bias the results due to the induction of metabolic stress? Inverse experiment with a KRAS inhibitor possible? A) granules are hardly visible. It is unclear, which quantification is fitting to which panel. B) NUPR1/G3BP1 western blot together with a control of ERK phosphorylation? Extent of NUPR1 induction correlated with the activity of canonical KRAS signaling?

Figure 4: High doxycycline concentration used. Again, some experiments with a selective KRAS inhibitor might support the experiments and might control the possible effects of doxycycline on the respiratory chain. Is the effects shown in figure 3B-D correlated to the KRAS signaling output? E-G: only one siRNA targeting sequence was used, which is below the current standard.

Figure 5: C-D: Corresponding western blots (p-ERK/pan-ERK etc.) from protein lysates at the endpoint of the treatment with ZZW-115?

Referee #3 (Comments on Novelty/Model System for Author):

Adequate Models used to elaborate on the underlying hypothesis

Referee #3 (Remarks for Author):

In this study, Santofimia-Castaño et al. studied the role of NUPR1 in pancreatic cancer regarding LLPS-mediated stress granule (SG) formation and its therapeutic usage in KrasG12D-dependent tumors. Considering the rising importance of LLPS in cancer and therapeutic approaches, this study holds significant potential. Authors have observed NUPR1-mediated stress granule formation in pancreatic cancer cells. They could genetically and pharmacologically reverse this phenotype. Significantly, KrasG12D mutation might increase NUPR1, therefore SGs too. On the other hand, NUPR1 overactivation can induce SG formation in pancreatic cancer independently of KrasG12D mutational load. Inhibition of NUPR1 increases Caspase-3 activation, thereby, cell death-related modalities, including increased LDH release.

Moreover, pharmacological targeting of NUPR1 via ZZW-115 inhibits SG formation and preneoplastic lesion formation in the KC model. In vitro, it again induces Caspase-3 and other apoptosis-related cell death mechanisms, showing its therapeutic relevance in KrasG12D mutated pancreatic cancer cells. This study is nicely conceptualized and driven by a logically designed experimental flow. However, there are some points that the authors would improve in this study.

In Figure 1., the authors tested the capacity of an IDP to induce LLPS with increasing concentrations of wild-type rNUPR1. They observed that rNUPR1 could undergo phase separation in a concentration-dependent manner (Figure 1A). However, mutations at different positions prevented phase separation of rNUPR1 (Figure 1B). These targets were also relevant to the designed inhibitor, ZZW-115 (Figure 1C-E). In the presence of this inhibitor, phase separation has been averted. This part of the study constructs an excellent base for the rest.

In Figure 2., the authors depicted the involvement of NUPR1 in SG formation in human and murine PDAC cells. Both pharmacologically and genetically depleted NUPR1 decreased SG formation significantly.

In Figure 3., the authors have analyzed the formation of SGs in inducible murine PDAC cells. Interestingly, they observe that SG formation may occur independently of Kras induction. Generated cells, including NUPR1-Flag, NUPR1A33Q/T68A-Flag, and GFP, have shown that SG formation could be visualized via the forms of NUPR1 by arsenite treatment. However, seeing the resolution of bar graphs in conditions where SGs are close to zero could be better. Additionally, having co-localization ratios of NUPR1 and G3BP1 would be nice to visualize the distribution of stress granules.

In Figure 4., the authors depicted cell death via FACS using Annexin V and PI staining. In Figure 4A, normalizing values according to their controls in bar graphs would be better. To understand whether cell death is mediated by Caspase3/7 activity, using zVAD to rescue the effect of ZZW-115 would be beneficial. Additionally, to know whether the double siRNA approach would have an additional impact on cancer cells, the researchers should combine siNUPR1 and siG3BP1 in cancer cells.

In Figure 5., the authors should classify ADM and PanIN lesions (low grade-high grade) in their KC model with or without treatments. This would also show the significance of NUPR1 and SGs in different preneoplastic lesions. When they start treatment, it will be necessary to see whether NUPR1 is expressed in KC mice via NUPR1 IHC. Additionally, it would be great to understand whether ZZW-115 leads to cell death in KC mice following treatment; authors should provide serum LDH, Amylase, and Lipase levels. The quantifications of Figures 5B-G should be provided.

In Figure 6., the authors have shown the expression profiles of NUPR1, G3BP1, p-EIF2a, and PABP following the ZZW-115 treatment of KC mice. Additionally, Figure 7 shows the increase of cleaved caspase-3 following ZZW-115 treatment. However,

previous results show that this treatment has reversed PanIN progression. So then, does apoptotic cell death occur in the healthy pancreata? The pictures show that cleaved caspase-3 has a higher expression profile in intact acinar cells. This proves the need for blood serum analyses, including previously mentioned parameters.

In Figure 8., ZZW-115 decreases the levels of SG in the KC mouse model. However, the stainings show multiple dots in the microenvironment of PanIN lesions. Did they provide PanIN or pancreas-specific NUPR1 and G3BP1 levels, or do graphs have whole dots, including the cellular component of the microenvironment? This point should be clarified to see whether ZZW-115 changes SGs in a cell-specific manner. Moreover, having multiple IHCs/Iifs showing the cells in the microenvironment would be helpful.

In Figure 9., ZZW-115's effect is quite visible. However, they should again provide detailed analyses of ADM and PanIN lesions. Additionally, including an IHC showing apoptosis levels in the tissues is necessary. On the other hand, the researchers should check the effect of ZZW-115 in the inflammation-driven carcinogenesis model using KC mice to strengthen their therapeutic approaches.

If authors could improve their manuscript in the line of mentioned points, the scope of their work will become much more relevant for the journal.

Referee #1 (Comments on Novelty/Model System for Author):

No suggestions to improve

Referee #1 (Remarks for Author):

In this manuscript Santofimia-Castaño et al demonstrate that the transcriptional regulators NUPR1 mediates the formation of the non-membranous organelles known as stress granules (SGs) in pancreatic cancer cells and pancreatic intraepithelial neoplasia (PanINs). The authors nicely demonstrate that NUPR1-dependent SG formation is mediated by the capacity of NUPR1 to undergo liquid-liquid phase separation (LLPS) via its intrinsically disordered region. A synthetic compound ZZW-115 that binds to and blocks residues critical to LLPS blocks SG formation. Pancreatic cancer is a very aggressive and therapy-resistant disease for which few targeted therapies exist. Enhanced SG formation by mutant KRAS has been shown to contribute the resistance of mutant KRAS cancer cells to cell stress, and more recently SGs were shown to play an important role in pancreatic tumorigenesis (PMID: 35674408). As such, SG biology and their targeting has important implications for pancreatic cancer therapeutics. The authors show that mutant KRAS upregulates NUPR1 expression and that genetic or pharmacological inhibition of NUPR1 impedes SG formation in vitro and in mouse models of PanINs. Notably, they show that ZZW-115 blocks SG formation and PanIN development and expansion. The findings of this study are novel, timely, and have important therapeutic implications. A couple of minor comments are listed below:

We express our gratitude to the reviewer for their encouraging comments about our work.

1.A schematic of the domains/structure of NUPR1 and the positioning of the IDR region and the A33Q, T68Q or A33Q/T68Q residues that are targeted by ZZW-115 would help to visualize/make clear the relevance of these residues to LLPS and how ZZW-115 may impair LLPS.

We thank the reviewer for this comment. The docking of ZZW-115 against NUPR1, utilizing fragments of various lengths encompassing the entire protein, has been detailed in our previous publication (PMID: 30920390). In such publication, Figure 1B illustrates how ZZW-115 aligns with the regions around Ala33 and Thr68 of NUPR1. We believe that presenting a similar figure would be deemed as self-plagiarism, and thus, we decided to refer the reviewer to the aforementioned figure for reference.

2.The data that NUPR1 inhibition or knockdown leads to SG inhibition, cell death, and impedes PanIN development and growth are very clear. However, NUPR1 has additional interactions/functions than the demonstrated binding to SG core proteins under stress and promoting SG formation. The authors cannot rule out that the observed effects are not due to such additional functions. This can be tested by rescuing SG formation through using synthetic constructs or hyperactive G3bp1 mutants (PMID: 32302571; PMID: 35674408). While the lack of these experiments does not detract from the novelty and significance of the findings, it should be discussed in the manuscript in the discussion section.

We consider this comment highly pertinent. Indeed, it cannot be ruled out that NUPR1 plays a crucial role in cell viability through its involvement in other multiprotein complexes. However, it is unambiguous the finding that NUPR1 is indispensable for SGs formation. While the overexpression of G3BP1 induces SGs formation, this phenomenon depends upon the presence of NUPR1, as inhibition of NUPR1 hampers the recovery of both SGs formation and cell viability with G3BP1 overexpression. Consequently, these findings underscore the pivotal role of NUPR1 in SGs formation, as also depicted in the new Figures EV1C and EV1D.

Referee #2 (Remarks for Author):

The authors observed that the stress-inducible nuclear protein NUPR1 is involved in the development of stress granules (SG). Under oncogenic stress, mediated by the expression of mutated KRAS, NUPR1 is essential for SG granule formation and the prevention of KRAS-driven caspase-mediated apoptosis. Consequently, treatment with the NUPR1 inhibitor ZZW-115, prevented/reduced PanIN formation and progression. This is a very important contribution opening the avenue for the chemoprevention of a deadly disease.

We sincerely appreciate this reviewer for her/his comments and for providing valuable feedback.

Considerations:

1) The effects of ZZW-115 on early ADM and PanIN development are impressive (Fig.5) as are the effects on later stages (Fig. 9). Are treatment data with ZZW-115 also available for more aggressive models or is the ZZW-115 effect dependent on tumor suppressors like CDKN2A or TP53? The authors treated the animals over longer periods with ZZW-115. Toxicity of the compound? Toxicity monitored? Was the on-target activity measured?

We regard both of these questions as highly relevant and clinically significant. While we did not employ the ZZW-115 compound in a more aggressive animal model of PDAC, our previous research (PMID: 24026351) utilized a genetic approach to investigate the impact of NUPR1 depletion on the highly aggressive PDAC mice model $Pdx1\text{-cre};LSL\text{-Kras}^{(G12D)};Ink4a/Arf^{(fl/fl)}$ or KIC. Cross-breeding KIC mice with $NUPR1^{-/-}$ mice revealed that approximately half of the animals did not develop PDAC at all, indicating that NUPR1 was mandatory for PDAC development in mice. These findings were published in 2014.

Concerning the potential toxicity of ZZW-115, we did not observe adverse effects during both the treatment period and an extended time of 90 days. Rigorous monitoring of animal weight and condition was consistently maintained throughout the study. Upon conclusion of the experiment, we meticulously collected and examined blood and organs. The appendix Figure S2 shows comprehensive histological analyses of the organs, the mice's weight trajectory throughout the experiment, and amylase activity in the blood, all aimed at confirming the absence of any detrimental effects on pancreatic physiology.

2) The authors argue that "inhibition of NUPR1-dependent SGs could be utilized as a synthetic lethality therapeutic strategy in mutated $Kras^{G12D}$ -dependent tumors". Considering the heterogeneity of the human disease, it would be important to provide some insight into a larger cohort of cellular models on how common and heterogeneous SG formation is in established human and murine pancreatic cancer cell lines. Is the number of stress granules correlating with the therapeutic efficacy of ZZW-115?

This is an interesting and clinically relevant question. We have induced the formation of SGs in 6 primary PDAC cells and observed the systematic induction of SGs. Viability measurements showed that all cells are sensitive to NUPR1 inhibition within a narrow IC_{50} range of ZZW-115. We can conclude that most of, if not all, PDAC cells induce SGs formation and are sensitive to NUPR1 inhibition. Results were presented in the new Figures EV1E and EV1F.

Minor points:

"Among these sensors, the serine/threonine kinases promote phosphorylation of eIF2 α , a main inductor of stress granules (SGs) formation, a membrane-less organelle (MLO) sensed to protect the

cell against the stress (McInerney et al, 2005)." There are several current reviews on SGs - a very detailed and critical is written by Glauninger et al. in Molecular Cell. Worth to cite?

We really appreciate this suggestion. We have cited this reference in the revised version of the manuscript.

Figure 2C: a NUPR1 knockdown control by western blot is needed. It would also be important to see protein expression of NUPR1 and G3BP1 by western blot, to control the arsenite-dependent induction. It is also unclear, how SGs are visualized in 2C. The y-axis is labeled as SGs per nucleus, which might be a vague description for the cytoplasmic SGs. Perhaps it is easier to follow the manuscript if the panels are in the reading order.

We appreciate these remarks. However, it is important to say that we have previously demonstrated the efficacy of the siRNA sequence by using Western-blot analyses (Figure S1 of PMID: 30451898); and by RT-qPCR (Figure 6 of PMID: 35869257). Consequently, we have cited these works in the Materials and Methods section of the revised version of the manuscript.

Concerning the levels of NUPR1 and G3BP1, we monitored protein expression either after treatment with arsenite alone, or in combination with ZZW-115. We found that G3BP1 levels did not change with the treatment, whereas there was a slight elevation in NUPR1 expression following arsenite treatment. This is consistent with NUPR1 being a stress protein rapidly upregulated upon stress induction. This observation aligns with the role of NUPR1 in driving SGs formation. These results are presented in the new Figure EV1B.

We also appreciate the two last comments of the reviewer. Consequently, we have added arrows to the images for easy identification of SGs. Additionally, we have restructured the description panel order in the text.

Mycoplasma testing protocol not reported.

This is an insightful observation. In our laboratory, regular mycoplasma testing is a mandatory practice, and it is carried out monthly for every cell line. The specific kit employed in our culture platform is *Lonza's MycoAlert® Mycoplasma Detection Assay*. We have incorporated this pertinent information in the updated version of the manuscript.

Figure 3: Doxycycline concentration too high? Might bias the results due to the induction of metabolic stress? Inverse experiment with a KRAS inhibitor possible? A) granules are hardly visible. It is unclear, which quantification is fitting to which panel. B) NUPR1/G3BP1 western blot together with a control of ERK phosphorylation? Extent of NUPR1 induction correlated with the activity of canonical KRAS signaling?

We find these inquiries interesting. We have adhered to established protocols outlined in our prior publications and those of our collaborators (PMID: 23761328, PMID: 34649604 and PMID: 35245687). However, to ensure that doxycycline treatment did not introduce any metabolic bias, we treated the cells with the specific Kras inhibitor MRXT1133 and assessed the mitochondrial activity by using the Seahorse device. Our findings conclusively demonstrated that doxycycline, at the given concentration, induced changes in mitochondrial activity corresponding to the activation on KRAS^{G12D} (as presented in figure EV2A). Remarkably, this increased activity was completely reversed by using the KRAS^{G12D} inhibitor MRXT1133. Therefore, the observed differences are attributable to Kras activation rather than doxycycline treatment. Furthermore, we employed the Kras inhibitor to investigate whether doxycycline treatment affected SGs formation. As illustrated in figure EV2C of the revised manuscript, our results unequivocally showed that the Kras inhibitor hampered SGs formation, corroborating the pivotal role of Kras activation in triggering SGs development.

Regarding points A and B, we agree on their significance. Therefore, we performed Western-blot analyses to evaluate Kras signaling levels following the enforced overexpression of NUPR1. Notably, there was no discernible increase in Kras signaling in the absence of doxycycline, as indicated by ERK phosphorylation. This finding confirms that NUPR1 overexpression induced SG formation independently of Kras signaling. Phospho-ERK exhibited robust activation in the presence of doxycycline as expected. However, this activation was constrained when SGs formation was induced as a consequence of the forced overexpression of NUPR1, as illustrated in the new Figure EV3C. The observed attenuation in Kras signalization in response to SGs formation held biological significance, providing a potential explanation for the observed cell growth arrest during cellular stress. This phenomenon could be attributed to the sequestration of a positive mediator such as G3BP1 into the formed SGs. Supporting this hypothesis, we observed that overexpression of G3BP1, under NUPR1-expressed conditions, but not in the absence of NUPR1, formed SGs in which it remained sequestered. These results collectively underscore the intricate interplay between SGs formation induced by NUPR1, G3BP1 sequestering, Kras signaling, and how the triggering of SGs formation is influencing the cellular responses to stress.

Figure 4: High doxycycline concentration used. Again, some experiments with a selective KRAS inhibitor might support the experiments and might control the possible effects of doxycycline on the respiratory chain. Is the effects shown in figure 3B-D correlated to the KRAS signaling out-put? E-G: only one siRNA targeting sequence was used, which is below the current standard.

As shown in figure EV2A, our findings indicate that doxycycline, at the concentration employed, did not impact the respiratory chain. Furthermore, we conducted a Caspase 3/7 activity test to confirm that ZZW-115-induced Caspase activity was contingent upon Kras activation (Figure EV4G). We assessed that ERK phosphorylation was activated by ZZW-115 treatment in a dose-dependent manner (Figure EV4H). This knocked-down of NUPR1 is paralleling the forced expression of the NUPR1. This additional dataset supports the conclusion that inhibiting NUPR1 does not result in the arrest of Kras signaling.

Regarding the reviewer's observation on the siRNA targeting sequence, we acknowledge the validity of question. However, it is crucial to note that in our case, this approach was employed to validate results obtained with the pharmacological inhibitor. Additionally, it is important to clarify that the siRNAs utilized in these experiments are commercial ones, encompassing five different sequences to ensure almost complete inhibition of expression. The revised manuscript incorporates RT-qPCR data demonstrating the inhibitory efficacy of the siRNA pool (Figure EV1A).

Figure 5: C-D: Corresponding western blots (p-ERK/pan-ERK etc.) from protein lysates at the endpoint of the treatment with ZZW-115?

We appreciate this suggestion. We have incorporated a Western-blot into the manuscript which illustrates the Kras signaling in the pancreatic tissue following the treatment (Figure 5H).

Referee #3 (Comments on Novelty/Model System for Author):

Adequate Models used to elaborate on the underlying hypothesis

Referee #3 (Remarks for Author):

In this study, Santofimia-Castaño et al. studied the role of NUPR1 in pancreatic cancer regarding LLPS-mediated stress granule (SG) formation and its therapeutic usage in KrasG12D-dependent tumors. Considering the rising importance of LLPS in cancer and therapeutic approaches, this study holds significant potential. Authors have observed NUPR1-mediated stress granule formation in

pancreatic cancer cells. They could genetically and pharmacologically reverse this phenotype. Significantly, KrasG12D.mutation might increase NUPR1, therefore SGs too. On the other hand, NUPR1 overactivation can induce SG formation in pancreatic cancer independently of KrasG12D mutational load. Inhibition of NUPR1 increases Caspase-3 activation, thereby, cell death-related modalities, including increased LDH release.

Moreover, pharmacological targeting of NUPR1 via ZZW-115 inhibits SG formation and preneoplastic lesion formation in the KC model. In vitro, it again induces Caspase-3 and other apoptosis-related cell death mechanisms, showing its therapeutic relevance in KrasG12D mutated pancreatic cancer cells. This study is nicely conceptualized and driven by a logically designed experimental flow. However, there are some points that the authors would improve in this study.

In Figure 1., the authors tested the capacity of an IDP to induce LLPS with increasing concentrations of wild-type rNUPR1. They observed that rNUPR1 could undergo phase separation in a concentration-dependent manner (Figure 1A). However, mutations at different positions prevented phase separation of rNUPR1 (Figure 1B). These targets were also relevant to the designed inhibitor, ZZW-115 (Figure 1C-E). In the presence of this inhibitor, phase separation has been averted. This part of the study constructs an excellent base for the rest.

We thank this reviewer for these comments.

In Figure 2., the authors depicted the involvement of NUPR1 in SG formation in human and murine PDAC cells. Both pharmacologically and genetically depleted NUPR1 decreased SG formation significantly.

We thank the reviewer for these comments.

In Figure 3., the authors have analyzed the formation of SGs in inducible murine PDAC cells. Interestingly, they observe that SG formation may occur independently of Kras induction. Generated cells, including NUPR1-Flag, NUPR1A33Q/T68A-Flag, and GFP, have shown that SG formation could be visualized via the forms of NUPR1 by arsenite treatment. However, seeing the resolution of bar graphs in conditions where SGs are close to zero could be better. Additionally, having co-localization ratios of NUPR1 and G3BP1 would be nice to visualize the distribution of stress granules.

We value the insightful comment and have carefully reconsidered our approach. Initially, we decided to follow the same resolution to maintain consistency across all treatments. However, in light of the reviewer's perspective, we recognize that this method might not be very informative. In response to this consideration, we have integrated the quantification of the co-localization between NUPR1-Flag and G3BP1 into the new Figure EV3A, therefore conveying more clearly our analysis.

In Figure 4., the authors depicted cell death via FACS using Annexin V and PI staining. In Figure 4A, normalizing values according to their controls in bar graphs would be better. To understand whether cell death is mediated by Caspase3/7 activity, using zVAD to rescue the effect of ZZW-115 would be beneficial. Additionally, to know whether the double siRNA approach would have an additional impact on cancer cells, the researchers should combine siNUPR1 and siG3BP1 in cancer cells.

This comment is very pertinent. In our previous works (PMID: 30920390 and PMID: 32446862), we already reported that ZZW-115 was capable of inducing cell death by apoptosis (among other kinds of cell death). We have performed experiment using ZVAD in the presence of ZZW-115 and the results are presented in the new figures EV4E and F of the revised manuscript. As expected, ZVAD treatment rescued partially the effect of the ZZW-115. Regarding the double inhibition, we have included the results in Figure 4E, F and G, although no additional effect was found.

In Figure 5., the authors should classify ADM and PanIN lesions (low grade-high grade) in their KC model with or without. This would also show the significance of NUPR1 and SGs in different

preneoplastic lesions. When they start treatment, it will be necessary to see whether NUPR1 is expressed in KC mice via NUPR1 IHC. Additionally, it would be great to understand whether ZZW-115 leads to cell death in KC mice following treatment; authors should provide serum LDH, Amylase, and Lipase levels. The quantifications of Figures 5B-G should be provided.

We express our gratitude to the reviewer for these valuable suggestions. In response, we have incorporated a sentence in the text highlighting that treatment with ZZW-115 effectively prevents the development of ADM and PanINs lesions, as evidenced by the absence of any such lesions in the treated mice. Furthermore, we have included the quantification of ADM and PanINs lesions in the new Figure EV5B of the revised manuscript.

Furthermore, we have incorporated immunofluorescence images capturing the pancreas of mice at the age of 5 weeks. At this stage, there is no significant expression of NUPR1, and no lesions have developed. The corresponding data are presented in the new Figure EV5A.

The discussion on the impact of ZZW-115 on KC mice is highly relevant and presents a compelling observation. Notably, our findings in KC mice indicate that cell death occurs exclusively in cells with activated Kras, as illustrated in Figure 7. Within the pancreas of these mice, only a limited number of cells undergo transformation, while the majority of the tissue maintains a healthy state. Addressing valuable feedback from this reviewer, we have enriched the revised manuscript with additional insights. Histological images of key organs, the mice's weight progression throughout the experiment, and amylase levels in the blood are now included in Appendix Figure S2. Furthermore, we have incorporated Western-blot for pERK, total ERK, pAKT, and the total amount of AKT proteins to enhance the quantification of their respective levels. Our analyses revealed a robust activation of pERK in KC pancreas, which was effectively inhibited by ZZW-115, resulting in the demise of Kras-activated cells during treatment. Remarkably, our findings also show that pAKT was activated in KC, albeit with lower sensitivity to ZZW-115 treatment. These compelling data are visually represented in Figure 5H of the revised manuscript. This comprehensive set of results not only strengthens the significance of our study but also provides a nuanced understanding of the differential effects of ZZW-115 on the molecular pathways involved.

In Figure 6., the authors have shown the expression profiles of NUPR1, G3BP1, p-EIF2a, and PABP following the ZZW-115 treatment of KC mice. Additionally, Figure 7 shows the increase of cleaved caspase-3 following ZZW-115 treatment. However, previous results show that this treatment has reversed PanIN progression. So then, does apoptotic cell death occur in the healthy pancreata? The pictures show that cleaved caspase-3 has a higher expression profile in intact acinar cells. This proves the need for blood serum analyses, including previously mentioned parameters.

This comment holds significant relevance. Specifically, the activation of caspases induced by ZZW-115 is observed exclusively in cells with activated Kras, sparing the unaffected healthy tissue. Consequently, the apoptotic cell death triggered by ZZW-115 was selectively confined to cells expressing mutated Kras, thereby stopping the progression and formation of pancreatic lesions. We have incorporated a serum analysis, revealing that ZZW-115 did not induce cell death in healthy acinar cells, in Appendix Figure S2.

In Figure 8., ZZW-115 decreases the levels of SG in the KC mouse model. However, the stainings show multiple dots in the microenvironment of PanIN lesions. Did they provide PanIN or pancreas-specific NUPR1 and G3BP1 levels, or do graphs have whole dots, including the cellular component of the microenvironment? This point should be clarified to see whether ZZW-115 changes SGs in a cell-specific manner. Moreover, having multiple IHCs/IFs showing the cells in the microenvironment would be helpful.

This comment is pertinent to our study. In response, the quantification process includes the dots presents in ADM and PanINs. We have addressed and clarified this aspect in the revised manuscript.

In Figure 9., ZZW-115's effect is quite visible. However, they should again provide detailed analyses of ADM and PanIN lesions. Additionally, including an IHC showing apoptosis levels in the tissues is necessary. On the other hand, the researchers should check the effect of ZZW-115 in the inflammation-driven carcinogenesis model using KC mice to strengthen their therapeutic approaches.

We fully endorse the reviewer's comment, and as a result, we have incorporated new images illustrating the pro-apoptotic effect of ZZW-115 (Figure 9C). Exploring carcinogenesis propelled by inflammation could indeed provide an excellent model for investigating the role of SGs in PDAC development. Notably, the protein NUPR1 is known to be overexpressed in pancreatitis, suggesting its potential protective role through the induction of SGs formation in the pancreas. However, in this particular study, our primary objective was to demonstrate, as a proof-of-concept, the critical role of NUPR1 in SGs formation in the context of Kras activation. The exploration of inflammation-driven carcinogenesis models is currently beyond the scope of this manuscript.

If authors could improve their manuscript in the line of mentioned points, the scope of their work will become much more relevant for the journal.

We are convinced that the comments made by this reviewer and the additional added experiments prompted by his/her important questions have guided us towards a significant improvement of the work.

9th Jan 2024

Dear Dr. Iovanna,

Thank you for submitting your revised manuscript. We have now received the reports from the referees who re-reviewed your manuscript, and as you will see below, they are supportive of publication. I will therefore be able to accept your manuscript once the following editorial points will be addressed:

1/ Manuscript text:

- Please accept previous changes, and only keep in track changes mode any new modification.
- Please remove the Graphical Abstract, the synopsis text and the footnotes from the manuscript text.
- Please provide up to 5 keywords.
- Materials and Methods:
 - o Cells: please indicate whether the cells were authenticated.
 - o Mice: please indicate the origin of the following strains: Pdx1-Cre and LSL-KrasG12D. Please provide the housing and husbandry conditions. Please provide age and gender of the mice at the time of experiments.
 - o Statistics: please include a statement on sample size, blinding, randomization, and inclusion/exclusion criteria.
 - o Please correct the checklist accordingly.
- Thank you for providing a Data Availability Section. Please note that only newly generated datasets, that haven't been previously published, should be listed here.
- Acknowledgements: please enter in the submission system all funding information listed in this section.
- Author contributions: CRediT has replaced the traditional author contributions section because it offers a systematic machine-readable author contributions format that allows for more effective research assessment. Please remove the Authors Contributions from the manuscript and use the free text boxes beneath each contributing author's name in our system to add specific details on the author's contribution. More information is available in our guide to authors.
- Please rename "Conflict of interest" / "Footnotes" to "Disclosure statement and competing interests". This section should follow the "Acknowledgements" section.

2/ Figures and Appendix:

- Please provide exact p values, not a range, in the figures or their legends.
- Figure legends:
 - Please note that a separate 'Data Information' section is required in the legends of figures EV 4e-g.
 - Please note that the legend for figure EV 4g is not labelled in the legend, however the corresponding figure panel for the same is labelled as EV 4g. This needs to be rectified.
 - Please define the annotated p values * in the legend of figure EV 1d as appropriate.
 - Please indicate the statistical test used for data analysis in the legends of figure 2d; EV 1d.
 - Please note that in figures EV 4c-d; there is a mismatch between the annotated p values in the figure legend and the annotated p values in the figure file that should be corrected.
 - Please note that information related to n is missing in the legends of figures 4c-d; 9b.
 - Although 'n' is provided, please describe the nature of entity for 'n' in the legends of figures 3a-b; 4b, e-g; 7; EV 1a, d; EV 2a; EV 3b; EV 4b-k.
 - Please note that the error bars are not defined in the legends of figures 2c-e; 4a; 6a-b; 7; 8; EV 1d; EV 2a; EV 3a.
 - Please note that the scale bar needs to be defined for figures EV 1c, e.
 - Please note that the white arrowheads are not defined in the legends of figures 2c-d; 3a; 6a-b. This needs to be rectified.
- Figure 5G is not referenced in the text, please add a callout accordingly.
- There are currently 7 supplementary tables uploaded as one docx file with tables, labeled as Supplementary Table 1, etc. Please make these tables Datasets, uploaded individually, and labeled Dataset EV1, etc. The callouts in the manuscript need to be updated accordingly.
- Appendix: please add page numbers in the table of content and in the file.
- Similarities were found between two images in Appendix Figure S2, lung samples. Please carefully check the figure composition.

3/ Thank you for providing Source Data. Please note that Source Data for main figures need to be uploaded as one folder per figure - all source data of one figure should be in one folder. Source Data for EV figures can be grouped and uploaded as one file/folder. Please also upload the completed SD checklist.

4/ Checklist:

Please make sure that the information provided in the Materials and Methods is also reported in the checklist, and complete the following sections: cell authentication, statistics, sample definition.

5/ Please provide 'The paper explained': EMBO Molecular Medicine articles are accompanied by a summary of the articles to

emphasize the major findings in the paper and their medical implications for the non-specialist reader. Please provide a draft summary of your article highlighting

6/ Synopsis:

- Please upload the synopsis text as an independent file. The synopsis should include a short stand first (maximum of 300 characters, including space) as well as 2-5 one-sentence bullet points that summarize the paper. Please write the bullet points to summarize the key NEW findings. They should be designed to be complementary to the abstract - i.e. not repeat the same text. We encourage inclusion of key acronyms and quantitative information (maximum of 30 words / bullet point). Please use the passive voice.
- Please provide the synopsis image as an independent png/tiff/jpeg file 550 px wide x 300-600 px wide, and make sure that the text remains legible.

7/ As part of the EMBO Publications transparent editorial process initiative (see our Editorial at <http://embomolmed.embopress.org/content/2/9/329>), EMBO Molecular Medicine will publish online a Review Process File (RPF) to accompany accepted manuscripts.

This file will be published in conjunction with your paper and will include the anonymous referee reports, your point-by-point response and all pertinent correspondence relating to the manuscript. Let us know whether you agree with the publication of the RPF.

I look forward to reading a new revised version of your manuscript as soon as possible.

Yours sincerely,

Lise Roth

***** Reviewer's comments *****

Referee #2 (Remarks for Author):

Congratulations to the authors! My concerns have been effectively addressed.

Referee #3 (Comments on Novelty/Model System for Author):

Adequate Models used to elaborate on the underlying hypothesis

Referee #3 (Remarks for Author):

The authors sufficiently responded to the raised concerns through providing new data, re-interpretation of already existing analysis and rephrasing parts of the discussion. I concur with the arguments in the rebuttal and congratulate the authors for this nice piece of work.

The authors addressed the editorial issues.

19th Jan 2024

Dear Dr. Iovanna,

Thank you for providing your revised files. While most issues are solved, there are still a few editorial concerns. Therefore, before I can accept your manuscript, please address the following issues:

1/ Materials and Methods:

Please note that I haven't been able to find the following information:

- Cells: please indicate whether the cells were authenticated.
- Statistics: please include a statement on sample size, blinding, and randomization, even if it was not done (for instance: "No blinding was done").

2/ Data Availability:

Please note that only newly generated datasets, that haven't been previously published, should be listed here. It appears the mass spectrometry proteomics data have been published before. Therefore, please indicate "This study includes no data deposited in external repositories." in this section, and refer to the published dataset in the manuscript text as follows: "Data ref: Smith et al, 2001" or "Data ref: NCBI Sequence Read Archive PRJNA342805, 2017". In the Reference list, data citations must be labeled with "[DATASET]".

Further instructions are available at:

.

3/ Figures:

- Please provide high resolution figures.
- Please correct the callouts for Datasets EV1-7 in the manuscript text.

4/ Checklist:

Please make sure that the information provided in the Materials and Methods is also reported in the checklist, and complete the following sections: cell authentication, statistics, sample definition.

5/ Synopsis:

Please note that the synopsis text should follow the following format:

- a short stand first (maximum of 300 characters, including space)
- 2-5 one-sentences bullet points that summarizes the paper (maximum of 30 words / bullet point).

I resized your synopsis image to 550 px wide x 265 px wide (see attached). Let us know if it is fine or if you would like to modify it (in particular regarding text legibility).

6/ Please let us know if you agree with the publication of the Review Process File. This file will be published in conjunction with your paper and will include the anonymous referee reports, your point-by-point response and all pertinent correspondence relating to the manuscript.

I look forward to reading a new revised version of your manuscript as soon as possible.

Yours sincerely,

Lise Roth

The authors addressed the remaining editorial issues.

25th Jan 2024

Dear Dr. Iovanna,

I am pleased to inform you that your manuscript is accepted for publication and is now being sent to our publisher to be included in the next available issue of EMBO Molecular Medicine!

With kind regards,

Lise Roth
